# Advancing intercalation strategies in layered hybrid perovskites by bringing together synthesis and simulations

Lin-Jie Yang [1,6], Wenye Xuan [2,3,6], Sara Henda[4], Shaoyang Wang[4], Sai Kiran Rajendran [4], David B. Cordes [1], David N. Miller [1,5], Alexandra M. Z. Slawin [1], Lethy Krishnan Jagadamma[4], Hamid Ohadi [4], Hsin-Yi Tiffany Chen [3], Matthew S. Dyer [2] & Julia L. Payne [1] ✉

Finding ways to modify the electronic structure of halide perovskites is desireable as they have applications in a variety of devices, from photovoltaics to LEDs. Additionally, designing functional materials can be facilitated through the use of computation. Here, we have used a combination of synthesis and computation to screen for intercalated, layered hybrid perovskites. 54 compositions were screened and five compounds containing intercalated halogens were prepared as single crystals. A further compound, which was not screened and contained IBr, was prepared. We computationally identified an intercalated layered hybrid perovskite with a low bandgap and prepared it as a thin-film. Through examination of single crystal X-ray diffraction data, we provide some design guidelines for intercalation. The conformational flexibility in the organic ammonium cation allows rotations in the carbon backbone which change upon intercalation. Optoelectronic properties were studied using photoluminescence spectroscopy and preliminary photovoltaic device testing.

Lead halide perovskites have received considerable attention due to their outstanding optoelectronic properties[1]. Unlike 3D perovskites, layered perovskites can accommodate larger organic cations, which allow further tunability of their crystal structures[2]. However, layered hybrid perovskites, with compositions $(DA)(MA)_{n-1}Pb_nI_{3n+1}$ or $(BA)_2(MA)_{n-1}Pb_nI_{3n+1}$ (where DA diammonium, MA methyl ammonium and BA butylammonium or another monoammonium cation), which have single inorganic layers (denoted as $n=1$) separated by organic ammonium cations, are unsuitable for use as light absorbers in photovoltaic devices, due to their large band gaps[3,4]. Although some layered perovskites have shown promising stability in moist atmospheres[5], the insulating organic ammonium cation limits the photovoltaic properties of this family of materials due to the large

quantum confinement and high exciton binding energy ($E_b$) as a result of the dielectric mismatch between the inorganic and organic layers[6].

In 1986, Maruyama et al. reported that small molecules, including 1-chloronapthalene, o-dichlorobenzene and hexane, could be reversibly intercalated into layered hybrid perovskites $(C_{10}H_{21}NH_3)_2CdCl_4$ and $(C_9H_{19}NH_3)_2PbI_4$[7]. To the best of our knowledge, this was the first report of intercalation in layered hybrid perovskites. However, in this study, single crystals of the intercalated compound were not obtained and only changes in unit cell parameters could be observed. Mitzi et al. then looked at the intercalation of $C_6H_6$ and $C_6F_6$ into $(C_6F_5C_2H_4NH_3)_2SnI_4$ and $(C_6H_5C_2H_4NH_3)_2SnI_4$, respectively[8]. In this instance, intercalation of $C_6F_6$ into $(C_6H_5C_2H_4NH_3)_2SnI_4$ only resulted in a 0.04 eV change in band gap, despite the distance between the

[1]School of Chemistry, University of St Andrews, North Haugh, St Andrews, Fife KY16 9ST, UK. [2]Department of Chemistry, University of Liverpool, Crown St, Liverpool, L69 7ZD and Materials Innovation Factory, University of Liverpool, 51 Oxford St, Liverpool, UK. [3]Department of Engineering and System Science, National Tsing Hua University, Hsinchu, Taiwan. [4]SUPA, School of Physics and Astronomy, University of St Andrews, North Haugh, St Andrews, Fife KY16 9SS, UK. [5]Energy Storage Research Group, School of Chemistry and Physics, Faculty of Science, Queensland University of Technology (QUT), Brisbane, QLD, Australia. [6]These authors contributed equally: Lin-Jie Yang, Wenye Xuan. ✉e-mail: jlp8@st-andrews.ac.uk

$[SnI_4]_\infty$ layers changing from 16.3 to 20.6 Å[8]. More recently, intercalation has played an important role in the processing of organic-inorganic metal halides, as solvents such as DMF, etc. have been postulated to intercalate between $PbI_2$ layers[9–11]. Nag has also looked at intercalation in a number of compounds, including $(BA)_2PbI_4$ (where BA butylammonium) and $(PEA)_2PbI_4$ (where PEA phenylethylammonium), but we note that no single crystal structures were obtained from single-crystal X-ray diffraction[12]. In this work, $(BA)_2PbI_4$ displayed two peaks in the photoluminescence spectrum, which was attributed to two different areas of the crystal (edge and terrace), which suggested electronic interactions between neighbouring $[PbI_4]_\infty$ layers[12]. When iodine was intercalated, only a single emission was observed in the photoluminescence, and this was found at higher energies[12]. In this study, the lower energy peak had been attributed to edge emission. This process was reversible. The same group then went to look at hexane intercalation into $(DEA)_2PbI_4$ (where DEA decyl ammonium), which again changed the PL emission from dual to single emission[12]. However, the intercalated molecules were prone to deintercalation, which precluded the growth of crystals suitable for single-crystal X-ray diffraction studies[12]. As a result, $(PEA)_2SnI_4·C_6F_6$, previously prepared by Mitzi et al. was investigated[8,12]. It also showed dual emission in the PL spectra, and like the other compounds, the low-energy PL emission disappeared upon intercalation of the $C_6F_6$ molecule[12]. To complete the study, Nag et al. also looked at intercalation in $(C_mH_{2m+1}NH_3)_2PbI_4$, where the length of the carbon chain was systematically varied[12]. As the carbon chain length increased, the PL went from dual emission to single emission, with the loss of the low-energy peak[12]. Karunadasa looked at the intercalation of $I_2$ into $(CH_3(CH_2)_5NH_3)_2PbI_4$ and the related compound containing a terminal alkyl iodide group, $(ICH_2(CH_2)_5NH_3)_2PbI_4$[13]. In these compounds, $I_2$ was only stable for a short time, and no single-crystal XRD could be obtained for either material, preventing full structural characterisation of these materials. We note that the intercalation of $I_2$ was found to be more stable in $(ICH_2(CH_2)_5NH_3)_2PbI_4·xI_2$ than $(CH_3(CH_2)_5NH_3)_2PbI_4·xI_2$[13]. However, the exciton binding energy for these compounds were reduced upon intercalation, with a value of 180 meV being reported for $(ICH_2(CH_2)_5NH_3)_2PbI_4·xI_2$[13]. The intercalation of DMSO and DMF into $(PEA-OH)PbBr_4$ (where $PEA-OH=HOC_6H_5(CH_2)_2NH_3^+$) has also been studied[14]. Here, the intercalation of DMSO was very stable, due to hydrogen bonds between the PEA-OH and DMSO, enabling its use as a photodetector[14]. However, the changes in electronic structure were small, and $(PEA-OH)PbBr_4·DMSO$ also had a short carrier lifetime[14]. It was also possible to intercalate DMF into $(PEA-OH)PbBr_4$, and both $(PEA-OH)PbBr_4·DMF$ and $(PEA-OH)PbBr_4·2DMF$ were reported[14]. Variable quantities of DMF could be intercalated, which led to mixed-phase materials being observed[14].

We recently reported that the intercalation of molecular bromine in an $n = 1$, layered hybrid perovskite, $[H_3N(CH_2)_6NH_3]PbBr_4$, could adjust both the crystal structure and electronic structure, forming $[H_3N(CH_2)_6NH_3]PbBr_4·Br_2$[15]. This resulted in the introduction of a new band between the valence band maximum (VBM) and conduction band minimum (CBM). The effective mass was also calculated to be reduced by two orders of magnitude, indicating that there is an enhanced mobility in $[H_3N(CH_2)_6NH_3]PbBr_4·Br_2$[15]. Although only one intercalated/deintercalated material was studied, the study used a combination of crystallography and computational work to show that halogen bonding is a key non-covalent interaction involved in the intercalation process. Intercalation also offers the possibility of tuning the optical properties of other $n = 1$ perovskites.

Here, in order to probe the intercalation of halogen molecules into layered hybrid perovskites in more detail, we computationally screened intercalation in the $[H_3N(CH_2)_mNH_3]PbX_4·X_2$ (where $m = 5–10$, $X$ = Cl, Br or I) family. To reduce the trial-and-error cost in the experimental approach, we started by predicting the stability and electronic structures of the intercalated layered hybrid perovskites with density-functional theory (DFT) calculations. Using a parallel experimental approach, six intercalated layered perovskites have been prepared and their structures characterised by single-crystal X-ray diffraction (SCXRD). Through this combined study, we have identified some design criteria which can be used in the preparation of stable, intercalated layered perovskites.

A selection of the intercalated layered perovskites were fabricated into highly oriented thin films, including some that could not be synthesised as single crystals. Variable temperature photoluminescence (PL) spectroscopy was used to study their optical properties. We find that halogen molecule intercalation adjusts the broadband PL emission, indicating that it is an alternative method to manipulate the quantum confinement of layered perovskites[13].

## Results and discussion

In order to probe which combinations of inorganic layers ($[PbX_4]_\infty$, where $X$ = Cl, Br or I), organic ammonium cations $[H_3N(CH_2)_mNH_3]^{2+}$ (which vary in the length of the diamine, where $m = 5–10$) and halogen molecules ($X_2$) were amenable to intercalation, we began by using DFT calculations to screen a series of parent $[H_3N(CH_2)_mNH_3]PbX_4$ and intercalated perovskites $[H_3N(CH_2)_mNH_3]PbX_4·X_2$, where $m = 5–10$ and $X$ = Cl, Br or I. By inspection of the computed crystal structures, which utilised $[H_3N(CH_2)_6NH_3]PbBr_4$ as a starting model, we defined three sets of structural parameters (Fig. 1a), which could change upon intercalation of a halogen molecule into a layered hybrid perovskite. The intercalated perovskite will exhibit equal distances (and therefore equal halogen bonds) between the halogen molecule and $[PbX_4]_\infty$ layers ($D_1 = D_2$), assuming that the halogen molecule will intercalate halfway between the inorganic layers. The difference between $D_1$ and $D_2$ is noted as $\Delta D$ (where $|\Delta D| = |D_1 - D_2|$). The angles $\theta_1$ and $\theta_2$ should be as close to 180° as possible if halogen bonding is present. The final point to consider is the flexibility in terms of the conformation of the $[H_3N(CH_2)_mNH_3]^{2+}$ cation, as $sp^3$ hybridised carbons are free to rotate so that the carbon chain may twist[16].

It is also worth noting that before intercalation, the distance between two apical halide ions in adjacent $[PbX_4]_\infty$ sheets ($D_h$) is highly tuneable and is influenced by two structural parameters: the perpendicular distance between two adjacent $[PbX_4]_\infty$ sheets ($D_L$) and the layer-shift factor ($L_s$ i.e. the shift between two inorganic layers with respect to one another). $D_h$ and $D_L$ are strongly influenced by the size of the $[H_3N(CH_2)_mNH_3]^{2+}$ cation, and in this work, only linear diammonium cations of different lengths were explored. The detailed structural parameters from our computational studies are given in Table S1 and Figs. S1–3.

As shown in Supplementary Figs. 1–3, the layered perovskites which use $[H_3N(CH_2)_mNH_3]^{2+}$ cations with $m = 6–8$ are predicted to intercalate halogen molecules between the inorganic layers, based on their structural parameters ($D_1$, $D_2$, $\theta_1$ and $\theta_2$). These $[H_3N(CH_2)_mNH_3]^{2+}$ cations are of the optimum length, as when the $[H_3N(CH_2)_mNH_3]^{2+}$ cations are too short ($m = 5$), the interlayer space cannot accommodate the smallest halogen molecule. This is reflected in the small bond angles, $\theta_1$ and $\theta_2$. For example, the predicted structure of $[H_3N(CH_2)_5NH_3]PbCl_4·I_2$ has a small $\theta_1$ (155.96°) and there is significant octahedral tilting in the predicted structure of $[H_3N(CH_2)_5NH_3]PbCl_4$ (Supplementary Fig. 4a). In addition, the $E_{binding}$ of $[H_3N(CH_2)_5NH_3]PbCl_4·I_2$ is positive, indicating that $I_2$ intercalation is not stable. In contrast, when the $[H_3N(CH_2)_mNH_3]^{2+}$ cations are too long ($m = 10$), the distances between the intercalated halogen and inorganic octahedra ($D_1$ and $D_2$) are extremely large ($|\Delta D| > 2.99$ Å) and indicate that the halogen molecule is only bonded to one $PbX_6$ octahedron and is disconnected from the octahedron in the adjacent layer, (Supplementary Fig. 4b). When $m = 9$, only the largest halogen molecules ($I_2$) can be intercalated and both halogen bonds are essentially equivalent in length ($|\Delta D| < 0.02$ Å) (Supplementary Fig. 3).

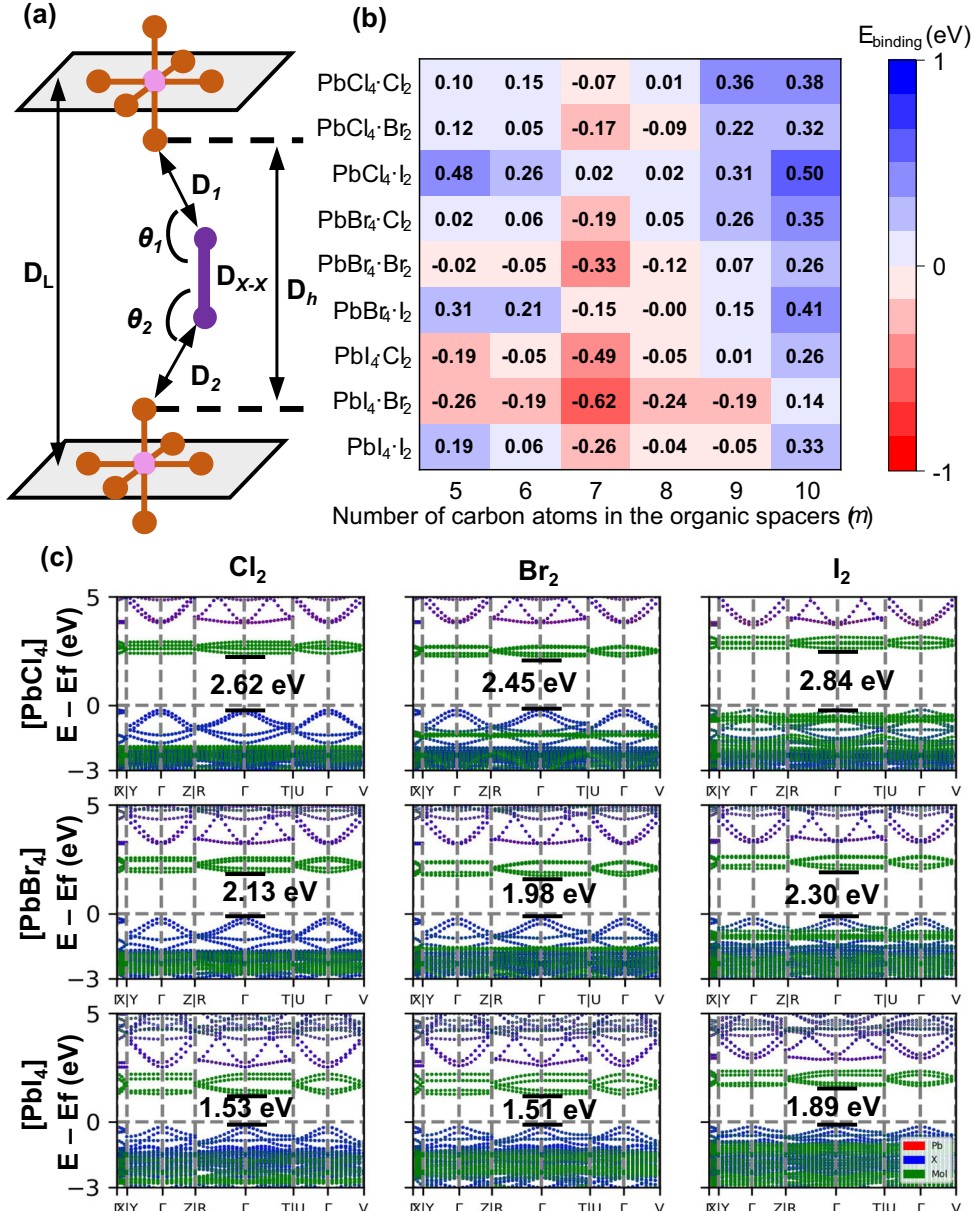

**Fig. 1 | Using DFT to probe the crystal structure and electronic structure.**
**a** Schematic illustration of the structural parameters used for computational
screening. *B*-site lead cation, *X*-site halogen anion (*X* = Cl, Br, I) and intercalated
halogen molecules are represented by pink, brown and purple spheres, respec-
tively. $D_h$ distance between the apical halide ions in adjacent $[PbX_4]_\infty$ sheets, $D_1$ and
$D_2$ distance between the intercalated halogen molecule and the halides in the
$[PbX_4]_\infty$ layers; $D_{X-X}$ bond lengths in the halogen molecule, $\theta_1$ and $\theta_2$ *X*-*X*-*X* angle. $D_L$
perpendicular distance between two adjacent $[PbX_4]_\infty$ sheets. **b** The binding energy
of the *m* = 5–10 family of intercalated perovskites, $([H_3N(CH_2)_mNH_3]PbX_4·X_2)$

calculated using DFT. Red indicates that intercalation is predicted to be energeti-
cally favourable. **c** Electronic structure of *m* = 6 family of intercalated perovskites,
$[H_3N(CH_2)_6NH_3]PbX_4·X_2$, calculated using hybrid functionals. The projection of Pb *p*
orbital, halide (*X*) *p* orbital of the perovskites and molecules (Mol) are denoted by
red, blue and green, respectively. The band gap values are also included. Please
note that as the conduction band is a mixture of orbital contributions from Pb (red)
and *X* (blue), the conduction band appears purple in the electronic structure
diagram.

The thermodynamic stability of the intercalated perovskites was
assessed by calculating the binding energy ($E_{binding}$) of the intercalated
halogen molecule as follows:

$$E_{binding} = (E_{intercalated} - (E_{parent} + E_{halogen}))/N_{molecules}$$

Where $E_{intercalated}$ represents the energy of the system after halogen
intercalation, $E_{parent}$ and $E_{halogen}$ are the energies of the parent layered
hybrid perovskites and halogen molecules that will be intercalated,
respectively. $N_{molecules}$ denotes the number of intercalated molecules
in the unit cell of the predicted structure. A negative $E_{binding}$ indicates
that the intercalation is an energetically favoured process and so the

calculations show which of the intercalated perovskites are more
stable than the isolated parent perovskite and halogen molecule
(Fig. 1b). A total of 54 compounds were screened, with eleven com-
positions predicted to exist. The influence of the length of the organic
ammonium cation was explored by looking at the trend in the $E_{binding}$
(Fig. 1b), which displays a minimum at *m* = 7. Experimental studies have
shown that the *m* = 7 $[H_3N(CH_2)_7NH_3]PbX_4$ parent structures are
unstable, and none could be prepared in single crystal form[17,18]. This
manifests itself in a high value of $E_{parent}$ for the *m* = 7 samples, which
leads to low values of $E_{binding}$. $[H_3N(CH_2)_mNH_3]PbX_4$ and odd *m* num-
bers (*m* = 5 or 9) also displayed high $E_{parent}$, as expected, as previous
crystallographic studies showed that they have low symmetry[17].

In general, Fig. 1b shows that for a particular value of $m$, the binding energy decreased from $Cl^-$ to $Br^-$ to $I^-$, suggesting that the halogen bond strength increases with heavier halides (as expected), and this can be attributed to the higher VBM of iodide-based perovskites.

To delve deeper into the electronic properties of intercalated perovskites, the electronic structure was calculated for both parent (Supplementary Fig. 5) and intercalated perovskites (Fig. 1c). All possible combinations based on the $[H_3N(CH_2)_6NH_3]^{2+}$ cation were studied, as single crystal structures of all parent $[H_3N(CH_2)_6NH_3]PbX_4$ perovskites have been previously reported[19–21]. Molecular orbital energy level diagrams were also calculated for each halogen molecule (Supplementary Fig. 6). In agreement with previous studies[22–24], the valence band character for both parent and intercalated perovskites was primarily from the halide $p$ orbitals and the VBM increased as the halide was changed from $Cl^-$ to $Br^-$ to $I^-$. Consequently, the band gap of intercalated perovskites decreased from $[H_3N(CH_2)_6NH_3]PbCl_4 \cdot X_2$, to $[H_3N(CH_2)_6NH_3]PbI_4 \cdot X_2$ for the same halogen molecule. Upon intercalation, a new band was introduced into the electronic structure. The conduction bands of the intercalated perovskites are from the $\sigma^*$ antibonding orbital of the halogen molecules, which arise from both the absolute energy of the halogen molecular orbitals and the upward and downward curving of the orbitals after intercalation[15]. We found that the band gap decreased in the order $[H_3N(CH_2)_mNH_3]PbX_4 \cdot I_2$, $[H_3N(CH_2)_mNH_3]PbX_4 \cdot Cl_2$ to $[H_3N(CH_2)_mNH_3]PbX_4 \cdot Br_2$ which differs from the order of the LUMO energies of the halogen molecules, as $Cl_2$ intercalation leads to a strong downwards curving dispersion of the LUMO states.

Through systematic computational studies of the intercalated perovskite, parent perovskite and halogen, combinations well-suited for photovoltaic applications were predicted by combining $[H_3N(CH_2)_mNH_3]PbI_4$ frameworks with $Br_2$. For example, $[H_3N(CH_2)_6NH_3]PbI_4 \cdot Br_2$ is shown to have a DFT calculated band gap of 1.51 eV. Within this family of compounds, both $[H_3N(CH_2)_6NH_3]PbI_4 \cdot Br_2$ and $[H_3N(CH_2)_7NH_3]PbI_4 \cdot Br_2$ have structural parameters that are within the optimum range for intercalation ($\theta_1$ and $\theta_2$ close to 180 ° and low $|\Delta D|$ = 0.02 Å, as given in Table S1). They also have large negative $E_{binding}$ (−0.19 eV for $[H_3N(CH_2)_6NH_3]PbI_4 \cdot Br_2$ and −0.62 eV for $[H_3N(CH_2)_7NH_3]PbI_4 \cdot Br_2$). Therefore, they were both selected as the most promising combinations for photovoltaic applications. During exploratory synthesis, single crystals were obtained for five of these $[H_3N(CH_2)_mNH_3]PbX_4 \cdot X_2$ ($m$ = 7–9) compounds with the heavier halides, bromide and iodide, as well as for $[H_3N(CH_2)_7NH_3]PbBr_4 \cdot IBr$, and these are highlighted in Table S2. However, the two most promising combinations mentioned above could not be synthesised in bulk form, due to the chemically favoured halide ion exchange between $Br_2$ and $I^-$, but the corresponding thin film form will be discussed later (vide infra).

Through the combination of computational screening and exploratory synthesis, the synthesis of five intercalated layered perovskites were attempted, and samples were obtained as single crystals (Table S2). We also successfully prepared a layered hybrid perovskite intercalating IBr. SCXRD was used to study the crystal structures of these six compounds. The resulting crystallographic details are given in Table S3, and the accompanying PXRD data are given in Figs. S7–12. The key structural parameters for a total of seven intercalated samples (including $[H_3N(CH_2)_6NH_3]PbBr_4 \cdot Br_2$)[15] are given in Table S4, and parts of their structure are displayed in Fig. 2a. Additionally, a comparison between the computational and experimental structural parameters have been given in Supplementary Fig. 13. We were able to synthesise crystals of $[H_3N(CH_2)_8NH_3]PbBr_4 \cdot Br_2$ but crystal quality was poor, and the structure showed large anisotropic displacement parameters for the bromine molecule and disorder of the $[H_3N(CH_2)_8NH_3]^{2+}$, which could indicate that $m$ = 8 is the 'upper limit' for stable intercalation of $Br_2$ molecules.

In agreement with our computational results, which showed that $E_{binding}$ displays a minimum at $m$ = 7, experiments showed that the $m$ = 7 family could intercalate halogen molecules. Their refined crystal structures are shown in Fig. 2b. Their experimental $D_L$ values were in a suitable range for the intercalation of three different halogen molecules ($Br_2$, $IBr$ and $I_2$). $D_L$ was also optimised by the change in carbon chain conformation upon intercalation. Although the three structures contain the same organic cation, the intercalation of different halogen molecules requires different conformations of the $m$ = 7 carbon chain. This in turn results in different layer-shift factors ($L_s$) for the structures: $[H_3N(CH_2)_7NH_3]PbBr_4 \cdot Br_2$ (0.19, 0.19); $H_3N(CH_2)_7NH_3]PbBr_4 \cdot IBr$ (0, 0) and $[H_3N(CH_2)_7NH_3]PbI_4 \cdot I_2$ (0.17, 0.17). In $[H_3N(CH_2)_7NH_3]PbBr_4 \cdot Br_2$, intercalation of $Br_2$ led to a rotation of one C−C bond and one C−N bond at the same end of the $[H_3N(CH_2)_7NH_3]^{2+}$, reducing its apparent length and making it structurally asymmetric. Therefore, in $[H_3N(CH_2)_7NH_3]PbBr_4 \cdot Br_2$, the length of the two halogen bonds which stem from the same bromine molecule were not identical ($D_1 \neq D_2$, $\theta_1 \neq \theta_2$) and non-centrosymmetric symmetry was adopted, unlike the centrosymmetric symmetry found in $[H_3N(CH_2)_6NH_3]PbBr_4 \cdot Br_2$[15]. Upon intercalation of the larger IBr molecule between the $[PbBr_4]_\infty$ layers, $[H_3N(CH_2)_7NH_3]^{2+}$ showed an extended chain confirmation with an '*all-trans*' form, as had been seen in $[H_3N(CH_2)_6NH_3]PbBr_4 \cdot Br_2$.

As shown in Fig. 2c, increasing the length of the $[H_3N(CH_2)_mNH_3]^{2+}$ cation results in larger $L_s$: $[H_3N(CH_2)_7NH_3]PbI_4 \cdot I_2$ (0.17, 0.17), $[H_3N(CH_2)_8NH_3]PbI_4 \cdot I_2$ (0.40, 0.46); and $[H_3N(CH_2)_9 NH_3]PbI_4 \cdot I_2$ (0.48, 0.48). This can be linked to the change in the carbon chain conformation as the shortest carbon chain ($m$ = 7) adopts a fully *all-trans* conformation, whilst the longer carbon chains ($m$ = 8 and 9) have room to rotate at the carbon−carbon/ carbon−nitrogen bonds. Therefore, in $[H_3N(CH_2)_7NH_3]PbI_4 \cdot I_2$, the full extension of the $m$ = 7 carbon chain resulted in the longest $D_L$ and the lowest $L_s$ of all $[H_3N(CH_2)_mNH_3]PbI_4 \cdot I_2$ structures, where $m$ = 7–9 (Supplementary Fig. 14).

Figure 2d shows a comparison between $[H_3N(CH_2)_8NH_3]PbBr_4 \cdot I_2$ and $[H_3N(CH_2)_8NH_3]PbI_2 \cdot I_2$ to probe the influence of the halide anions in the inorganic layer. The two structures have different $L_s$: $[H_3N(CH_2)_8NH_3]PbBr_4 \cdot I_2$ (0.06, 0.25) and $[H_3N(CH_2)_8NH_3]PbI_4 \cdot I_2$ (0.40, 0.46). The $[H_3N(CH_2)_8NH_3]^{2+}$ cation is too long in the all-*trans* conformation for iodine intercalation into $[H_3N(CH_2)_8NH_3]PbI_4$. In order to reduce $D_h$ in $[H_3N(CH_2)_8NH_3]PbI_4 \cdot I_2$, one C−N bond is rotated and the whole $[H_3N(CH_2)_8NH_3]^{2+}$ chain is tilted with respect to the inorganic layers. However, when the framework is changed to a bromide based $[H_3N(CH_2)_mNH_3]PbBr_4$, the conformation of the $[H_3N(CH_2)_8NH_3]^{2+}$ cation changes more drastically with a symmetrical rotation of the C−N bond at both of ends of $[H_3N(CH_2)_8NH_3]^{2+}$, reducing both $D_h$ and the resulting halogen bond length. According to Table S4, in $[H_3N(CH_2)_8NH_3]PbBr_4 \cdot I_2$, the halogen bond is between $Br^-$ and $I_2$ with the bond lengths of 3.2532(9) and 3.2567(9) Å which are shorter than the halogen bonds between $I^-$ and $I_2$ in $[H_3N(CH_2)_8NH_3] PbI_4 \cdot I_2$ (3.404(4) and 3.370(4) Å).

The distortion of $PbX_6$ octahedra before or after halogen intercalation must also be considered, as it has been linked to optoelectronic properties such as strong self-trapped exciton (STE) emission[25–27]. Table S5 shows that no significant Pb-$X$ bond length distortions were observed ($\Delta d \sim 10^{-6}$, Fig. 3a, b) either before or after intercalation, unlike the (110)-oriented layered perovskites[25,27–29]. There is essentially no distortion of $PbX_6$ octahedra when there is no size mismatch between the halide ion in the $[PbX_4]_\infty$ layers and the halogen atom from the intercalated molecule.

Equatorial Pb−$X$−Pb angles and equatorial Pb−Pb distances were used to monitor the inter-octahedral distortions within the inorganic layers (Fig. 3c and Table S5)[30]. The most significant change in inter-octahedral and individual octahedral distortion occurred upon intercalation of $I_2$ into $[H_3N(CH_2)_8NH_3]PbBr_4$ (Fig. 3a, d and Supplementary Fig. 15). Here, the $[PbBr_4]_\infty$ sheets expanded along the in-plane

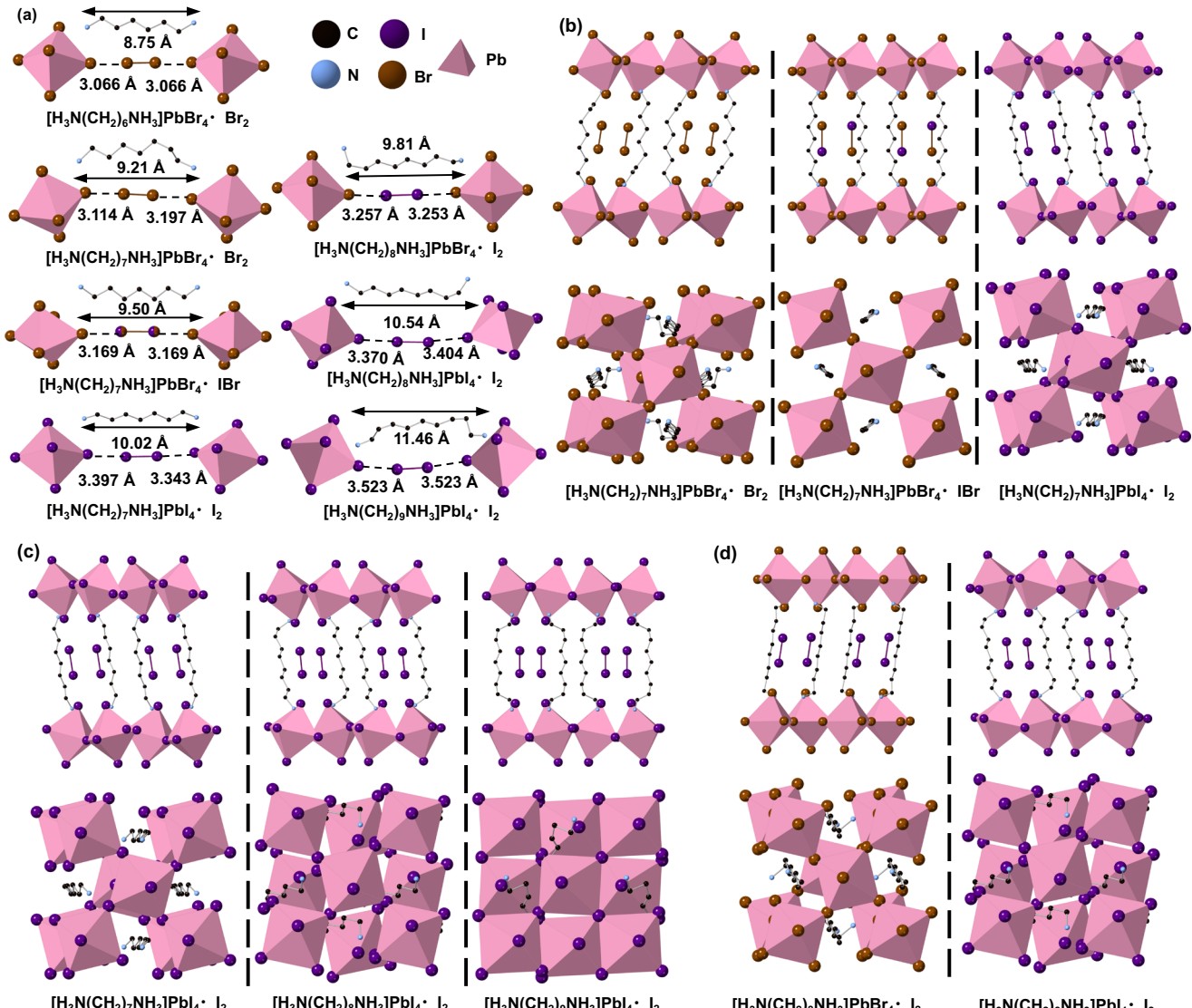

**Fig. 2 | Crystal structures of intercalated materials obtained from single-crystal X-ray diffraction data. a** Part of the crystal structure of seven intercalated perovskites ([H₃N(CH₂)ₘNH₃]Pb$X_2$·$X_2$). I, Br, N and C atoms are represented by purple, brown, blue and black spheres, respectively, whilst the Pb-centred polyhedra are shown in pink. Hydrogen atoms are omitted for clarity. The linear 'length' of the [H₃N(CH₂)ₘNH₃]²⁺ cation is labelled. **b** Two views of the crystal structures for all three $m = 7$ samples [H₃N(CH₂)₇NH₃]PbBr₄·Br₂, [H₃N(CH₂)₇NH₃]PbBr₄·IBr and [H₃N(CH₂)₇NH₃]PbI₄·I₂. **c** Two views of the crystal structure for all three [H₃N(CH₂)ₘNH₃]PbI₄·I₂ ($m = 7$, 8 and 9) intercalated samples. **d** Two views of the crystal structure for [H₃N(CH₂)₈NH₃]PbBr₄·I₂ and [H₃N(CH₂)₈NH₃]PbI₄·I₂ where the halide in the inorganic layers is different.

directions, allowing more space for the intercalation of the larger I₂ molecules. This indicated that the size mismatch between the halides in the inorganic layers and intercalated molecules influences octahedral/inter-octahedral distortion. Therefore, smaller halogens can be intercalated into [H₃N(CH₂)ₘNH₃]Pb$X_4$ frameworks, which contain larger halide ions, without distorting the lead halide frameworks. As such, if we could obtain SCXRD data of [H₃N(CH₂)₆NH₃]PbI₄·Br₂, we would expect the equatorial Pb–$X$–Pb angles and Pb–Pb distances to be different to those in [H₃N(CH₂)₆NH₃]PbI₄.

Figure 3e indicates that equatorial Pb–$X$–Pb angles and equatorial Pb–Pb distances increase with $m$ when it is the only variable. This can be linked to the area taken up by the [H₃N(CH₂)ₘNH₃]²⁺ cation, (i.e. the cross-sectional area of the cations) as the narrower 'all-*trans*' and 'over-stretched' conformations found in [H₃N(CH₂)₆NH₃]PbBr₄·Br₂ and [H₃N(CH₂)₇NH₃]PbI₄·I₂ enables the [H₃N(CH₂)ₘNH₃]²⁺ cation to penetrate further into the inorganic layer, shortening the equatorial Pb–$X$–Pb angles and Pb–Pb distances through enhanced hydrogen bonding interactions. Similar observations have been made in the literature[2,31].

In summary, the conformational flexibility of the [H₃N(CH₂)ₘNH₃]²⁺ cation, halogen bond length (**$D_1$**, **$D_2$**), $X$–$X$–$X$ bond angles (between halogen ions/atoms, **$\theta_1$** and **$\theta_2$**) and halide-halide distance can be used to determine whether intercalation in hybrid perovskites is possible. Size mismatch between the halogen atoms in the intercalated molecule and the halide ions in the [PbX₄]∞ layers, which contain smaller halides, also induces an expansion of the inorganic layers and distorts the octahedra, and as a result, these materials exhibit band gaps that may be potentially useful in optoelectronics.

Thermogravimetric analysis (TGA) was carried out on [H₃N(CH₂)ₘNH₃]Pb$X_4$·$X_2$ (Table S6 and Supplementary Fig. 16), to assess the thermal stability. The optical band gaps of [H₃N(CH₂)ₘNH₃]Pb$X_4$·$X_2$ were also assessed using diffuse reflectance UV-Vis spectroscopy (Supplementary Fig. 17 and Supplementary Table 7). Of the bulk samples synthesised, [H₃N(CH₂)₇NH₃]PbI₄·I₂ had the lowest band gap (1.77 eV) and the most promising thermal stability. In contrast, [H₃N(CH₂)₇NH₃]PbBr₄·Br₂ had a band gap of 2.39 eV and lower thermal stability.

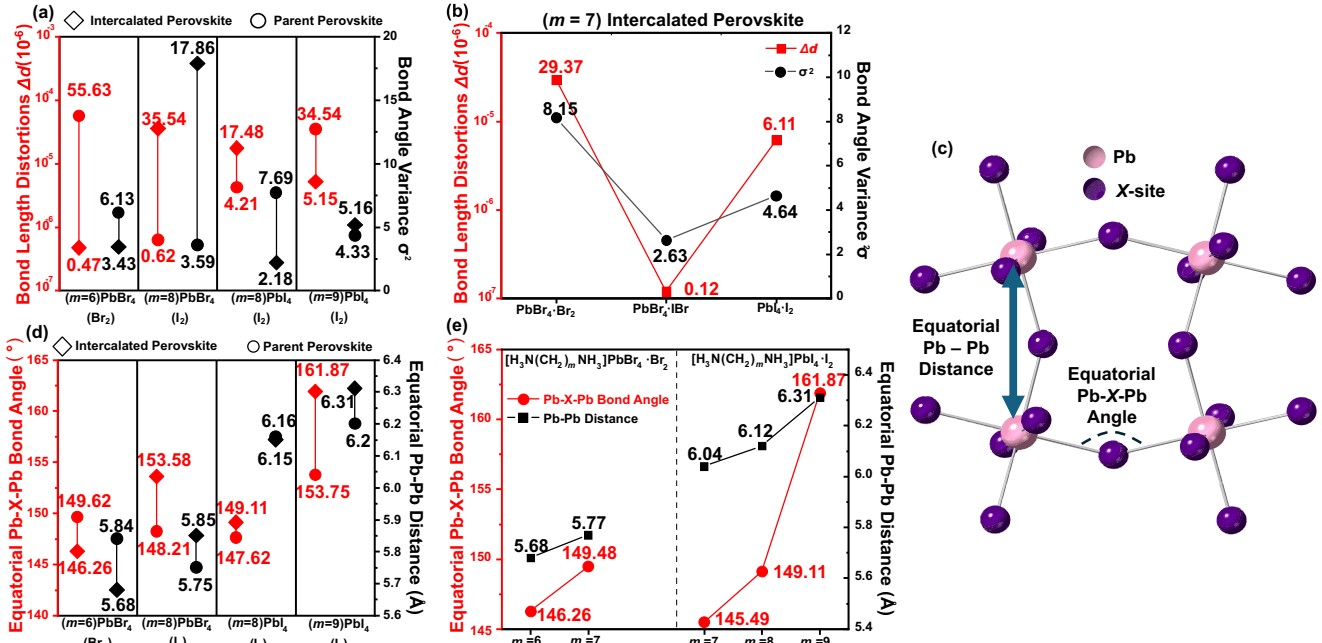

**Fig. 3 | Variation of octahedral distortion upon intercalation. a** Comparison of individual octahedral distortion parameters $\Delta d$ (scatter plot with reference to left y-axis) and $\sigma^2$ (scatter plot with reference to right y-axis) for parent[17,19,21,30,67] and intercalated structures ($m = 6, 8, 9$). **b** Comparisons of octahedral distortion for three $m = 7$ intercalated perovskites. **c** Schematic of two inter-octahedral distortion parameters: equatorial Pb–$X$–Pb angle and equatorial Pb–Pb distance. Pb is shown in pink and $X$ in purple. **d** Comparisons of inter-octahedral distortion parameters equatorial Pb–$X$–Pb angle (scatter plot with reference to left y-axis) and equatorial Pb–Pb distance (scatter plot with reference to right y-axis) before and after halogen intercalation. **e** Comparison of inter-octahedral distortions between the intercalated perovskites which differ in $m$ numbers.

In order to explore the preparation of intercalated layered perovskites, which had been predicted to exist but could not be formed as single crystals, such as $[H_3N(CH_2)_6NH_3]PbI_4 \cdot Br_2$, the fabrication of a selection of intercalated compounds as thin films was explored. This also allowed us to assess optoelectronic properties and the possibility of fabricating the intercalated layered perovskites into devices. We prepared thin films of the intercalated materials via a range of spin-coating methods (see Figs. S18–20). Firstly, we targeted the synthesis of $[H_3N(CH_2)_8NH_3]PbI_4 \cdot I_2$ (Fig. 4a) which could be prepared in bulk form, but the resulting thin film had a poor surface coverage (Fig. 4b). A second film fabrication method was tested which involved the post-synthetic intercalation of $I_2$ into $[H_3N(CH_2)_8NH_3]PbI_4$ films (Fig. 4c and Supplementary Fig. 21). PXRD (Fig. 4d) showed that the parent films could intercalate the desired halogen and that the same structures could be obtained as those from single-crystal XRD. The out-of-plane orientation (denoted as 001) of the parent perovskite thin film was maintained after intercalation. The surface coverage and uniformity of the films significantly improved (Fig. 4e, f). However, this post-synthetic method still didn't facilitate the preparation of thin film samples with the $m = 7$ $[H_3N(CH_2)_mNH_3]^{2+}$ cation, presumably due to the fact that the parent perovskites are unstable. Thin films of $[H_3N(CH_2)_8NH_3]PbBr_4 \cdot I_2$ cannot be prepared by post-synthetic intercalation, as the change in volume upon intercalation led to a degradation in film quality. Table S8 clearly outlines which perovskites are predicted and whether they can be synthesised as single crystals or by post-synthetic modification of thin films.

As computational studies predicted that $[H_3N(CH_2)_6NH_3]PbI_4 \cdot Br_2$ will have a low band gap, and we were unable to synthesise this material in bulk form, we decided to try and prepare this material in thin film form, using post-synthetic intercalation. Low-wavenumber Raman spectroscopy showed that extra bands appeared in the Raman spectrum of $[H_3N(CH_2)_6NH_3]PbI_4 \cdot Br_2$, in the region expected for solid $Br_2$, providing evidence for $Br_2$ intercalation (Supplementary Figs. 22, 23)[15,32,33]. The band gap obtained for this sample (Supplementary Fig.

24) is 2.05 eV and is lower than both the pure $[H_3N(CH_2)_6NH_3]PbBr_4 \cdot Br_2$ we reported previously (2.15 eV)[15] and $[H_3N(CH_2)_6NH_3]PbI_4$. This provides some evidence to show that intercalated bromine molecules and $X$-site iodide ions co-exist in the sample ($y < 4$ and $x > 0$). Therefore, as we were unable to determine the chemical composition of the film, this sample was denoted as $[H_3N(CH_2)_6NH_3]PbI_{4-y}Br_y \cdot xBr_2$, where $0 \le y \le 4$ and $0 \le x \le 1$. Diffuse reflectance UV-visible spectroscopy, PXRD and images of thin films provided some evidence for halide ion exchange in this sample (Supplementary Figs. 24–26 and Fig. 4g). Halide ion exchange hindered our attempts to prepare bulk samples of $[H_3N(CH_2)_6NH_3]PbI_4 \cdot Br_2$ and $[H_3N(CH_2)_7NH_3]PbI_4 \cdot Br_2$, both of which were predicted to be promising photovoltaic materials. A more detailed understanding of the competition between halide ion exchange and halogen intercalation is required, taking into consideration the quantity of $Br_2$ that the sample is exposed to and the reaction temperature. However, we note that Karundasa et al. found that halide ion exchange is chemically favoured and could occur by exposing the 3D perovskite $CH_3NH_3PbI_3$ to $Br_2$ or $Cl_2$ vapour, to make $CH_3NH_3PbBr_3$ and $CH_3NH_3PbCl_3$[34,35].

PL spectroscopy at 4 K was carried out on crystals of all seven crystalline intercalated perovskites and three parent perovskites $[H_3N(CH_2)_6NH_3]PbBr_4$, $[H_3N(CH_2)_8NH_3]PbBr_4$ and $[H_3N(CH_2)_8NH_3]PbI_4$, in order to probe their optoelectronic properties.

The parent perovskite crystal shows a double band emission under ultraviolet (415 nm) illumination—a narrow-band exciton green emission (Supplementary Fig. 27) and the broadband exciton red emission (Supplementary Fig. 28)[36]. The narrow-band emission is attributed to the free exciton (FE) that occurs at the band gap[37]. Several theories have been proposed for the presence of broadband emission in hybrid perovskites. These include extrinsic factors such as precursor stoichiometry[38] and edge states within perovskite crystals[39,40], low-lying trap states caused by defects in the crystal structure[41] and self-trapped excitons (STEs)[42]. Since these properties in parent perovskites are widely reported, in this paper, we focus our attention on the

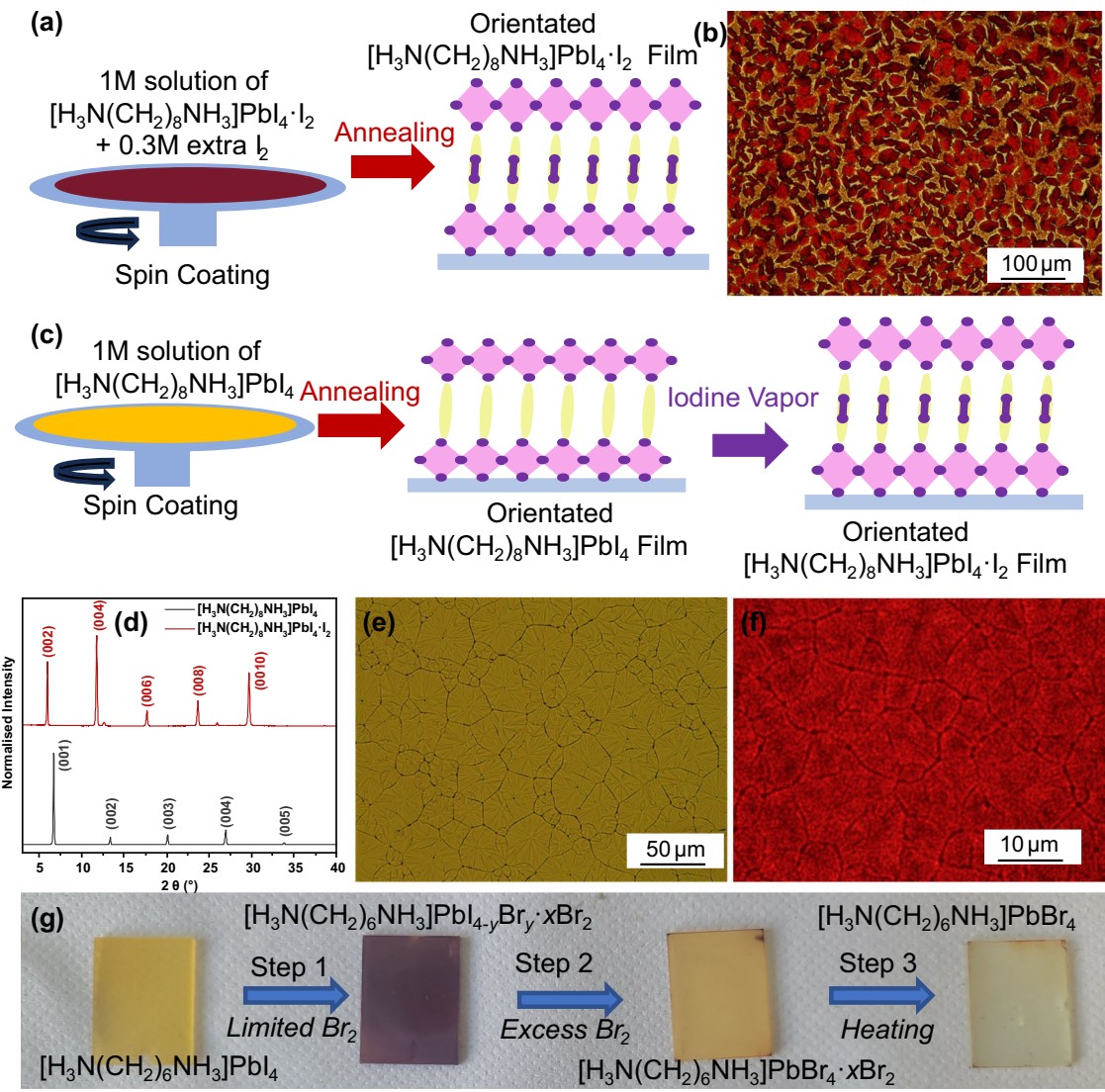

**Fig. 4 | Optimisation of film fabrication methods. a** Schematic showing the direct spin-coating method. **b** Images of thin films from optical microscopy studies of [H₃N(CH₂)₈NH₃]PbI₄·I₂ when prepared using the direct spin-coating method. **c** Schematic showing the process of the post-synthetic intercalation method. **d** Corresponding PXRD patterns of [H₃N(CH₂)₈NH₃]PbI₄·I₂ and [H₃N(CH₂)₈NH₃] PbI₄·I₂ thin films. Images of thin films from optical microscopy studies of **e** [H₃N(CH₂)₈NH₃]PbI₄ prepared using the direct spin-coating method and **f** [H₃N(CH₂)₈NH₃]PbI₄·I₂ prepared using the post-synthetic intercalation method. **g** Br₂ intercalation and bromide ion exchange process in a [H₃N(CH₂)₆NH₃]PbI₄ thin film prepared via a post-synthetic Br₂ intercalation method.

intercalated perovskites and the trends in emission with change in carbon chain length (*m* numbers) and intercalating halogens.

Initially, we look at the influence of varying the carbon chain length on the PL emission of intercalated perovskites. Similar to the parent perovskites, all [H₃N(CH₂)ₘNH₃]PbI₄·I₂ (*m* = 7, 8, 9) samples exhibit double emission under 415 nm ultraviolet illumination: a high intensity, green FE emission peak (Fig. 5a) and a low intensity broadband red emission peak (Fig. 5b). Notably, at room temperature, the FE emissions of [H₃N(CH₂)ₘNH₃]PbI₄·I₂ adhere to the same sequence (*m* = 8 <*m* = 7 and <*m* = 9), in terms of peak position (wavelength) of their parent perovskites[17]. In the intercalated perovskite, the FE emission peaks still originate from the conduction band of the parent perovskite. This trend agrees with the fact that there is only a small change in octahedral distortion before and after iodine intercalation. Previously, the shortest wavelength of the green FE emission was found for *m* = 8 and was attributed to the smallest equatorial Pb-I-Pb angle[17]. However, based on our work and the results in Fig. 3e, we suggest that more structural descriptors, such as the smallest **L_s** of the *m* = 7 family, should be used in order to draw more meaningful conclusions[43]. Supplementary Fig. 27a shows a small shift in emission

from the parent and intercalated perovskite crystals of the bright FE and a weak shoulder attributed to biexcitons at 4 K. At room temperatures, due to thermal broadening, the FE emission has a larger FWHM in both crystals and thin films of the parent and intercalated perovskites with only small differences (Supplementary Fig. 27b, c). In agreement with the earlier discussion on octahedral distortion, [H₃N(CH₂)₈NH₃]PbI₄·I₂ exhibited similar green emissions at the same wavelength (485 nm) as [H₃N(CH₂)₈NH₃]PbI₄, and displayed comparable peak shapes.

The large width of the broadband emission can be linked to the antibonding nature of the intercalated CBM and results from intense scattering with the phonons, leading to strong moving barriers of charge carriers. Interestingly, the broadband red emission of the intercalated perovskites (Fig. 5b) exhibits a different sequence in the wavelength of the longer emission, corresponding to a smaller band gap, with increasing *m* (*m* = 7 <*m* = 8 <*m* = 9). This sequence contradicts the sequence for band gaps determined at room temperature from UV-Visible spectroscopy, *m* = 7 <*m* = 9 <*m* = 8, which suggests that attributing the broadband emission solely to the new CBM emission oversimplifies this phenomenon. The broadband peak position is

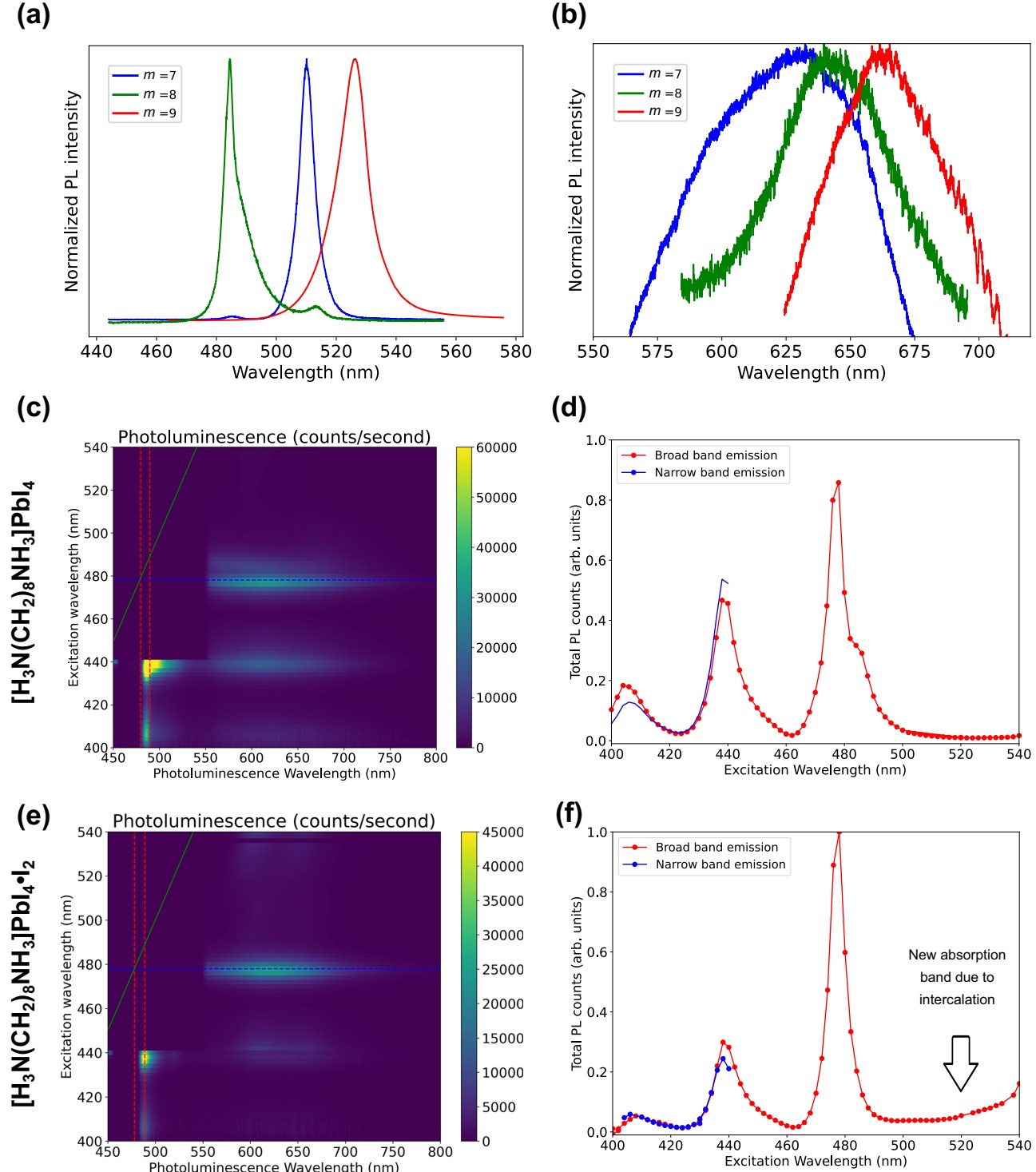

**Fig. 5 | Photoluminescence studies of intercalated materials. a** Comparison of the sharp photoluminescence peak of [H$_3$N(CH$_2$)$_7$NH$_3$]PbI$_4$·I$_2$, [H$_3$N(CH$_2$)$_8$NH$_3$]PbI$_4$·I$_2$ and [H$_3$N(CH$_2$)$_9$NH$_3$]PbI$_4$·I$_2$ crystals at 4 K; **b** Comparison of the broad pho-toluminescence peak of [H$_3$N(CH$_2$)$_7$NH$_3$]PbI$_4$·I$_2$, [H$_3$N(CH$_2$)$_8$NH$_3$]PbI$_4$·I$_2$ and [H$_3$N(CH$_2$)$_9$NH$_3$]PbI$_4$·I$_2$ crystals at 4 K; **c** Photoluminescence excitation spectra of [H$_3$N(CH$_2$)$_8$NH$_3$]PbI$_4$ measured at 4 K. **d** Combined emission spectra, showing contributions from both narrow-band and broad-band emissions of [H$_3$N(CH$_2$)$_8$NH$_3$]PbI$_4$, recorded with excitation wavelengths ranging from 400 to 540 nm. **e** Photoluminescence excitation spectra of [H$_3$N(CH$_2$)$_8$NH$_3$]PbI$_4$·I$_2$ mea-sured at 4 K. **f** Combined emission spectra, showing contributions from both narrow-band and broad-band emissions of [H$_3$N(CH$_2$)$_8$NH$_3$]PbI$_4$·I$_2$, recorded with excitation wavelengths ranging from 400 to 540 nm.

also correlated with the type of intercalated molecules. In Supple-mentary Fig. 28a, b, the effect of iodine intercalation on [H$_3$N(CH$_2$)$_8$NH$_3$]PbI$_4$ and [H$_3$N(CH$_2$)$_8$NH$_3$]PbBr$_4$ are observed. Broad-band emissions with a full-width at half maximum (FWHM) exceeding 100 nm were observed at different wavelengths (660 nm for [H$_3$N(CH$_2$)$_8$NH$_3$]PbI$_4$ and 642 nm for [H$_3$N(CH$_2$)$_8$NH$_3$]PbI$_4$·I$_2$; 600 nm for [H$_3$N(CH$_2$)$_8$NH$_3$]PbBr$_4$ and 560 nm for [H$_3$N(CH$_2$)$_8$NH$_3$]PbBr$_4$·I$_2$). This blueshift in broadband emission upon iodine intercalation con-trasts with the behaviour observed for bromine intercalation in [H$_3$N(CH$_2$)$_6$NH$_3$]PbBr$_4$ (520 nm) and [H$_3$N(CH$_2$)$_6$NH$_3$]PbBr$_4$·Br$_2$ (565 nm), shown in Supplementary Fig. 28c. Additionally, unlike iodine intercalated samples, broadband emission in bromine intercalated

perovskites does not exhibit sensitivity to changes in the $m$ numbers. The broadband emission peaks of $[H_3N(CH_2)_6NH_3]PbBr_4 \cdot Br_2$ and $[H_3N(CH_2)_7NH_3]PbBr_4 \cdot Br_2$ are comparable, both have peaks at 565 nm (Supplementary Fig. 29a). This can be attributed to their similar $L_s$ – (0.19,0.19) for $[H_3N(CH_2)_7NH_3]PbBr_4 \cdot Br_2$ and (0.13, 0.13) for $[H_3N(CH_2)_6NH_3]PbBr_4 \cdot Br_2$[15]. However, intercalating different molecules (IBr and $Br_2$) into the same parent structure ($[H_3N(CH_2)_7NH_3]PbBr_4$) also influenced the peak position of broadband emissions (Supplementary Fig. 29b).

To further investigate the origin of the broadband emission, variable temperature PL spectroscopy was conducted on crystalline samples of both $[H_3N(CH_2)_8NH_3]PbI_4$ and $[H_3N(CH_2)_8NH_3]PbI_4 \cdot I_2$ (Supplementary Fig. 30a and 30b) from 4 K to room temperature. The intensity of the broad peak decreases upon increasing temperature for both samples. However, this peak disappears at temperatures about 70 K in $[H_3N(CH_2)_8NH_3]PbI_4$, but in $[H_3N(CH_2)_8NH_3]PbI_4 \cdot I_2$ it is present up to 120 K. This result shows that the intensity of the broad peak is a thermally activated feature. Many studies have reported the temperature dependence of the STE emission[36]. At low temperatures, the thermal activation energy is lower than the trap energy, and broadband emission is observed from the STE. The antibonding nature of the inserted intercalated band in $[H_3N(CH_2)_8NH_3]PbI_4 \cdot I_2$, is non-dispersive and leads to a weak potential for charge carriers to relax to the conduction band minimum. This results in intense scattering with the phonons, hence emission from the intercalated band was non-detectable until 120 K or lower.

In order to unravel the nature of the species with red broadband emission in the intercalated perovskite, we performed a comprehensive photoluminescence excitation (PLE) experiment on the $[H_3N(CH_2)_8NH_3]PbI_4$ (Fig. 5c, d) and $[H_3N(CH_2)_8NH_3]PbI_4 \cdot I_2$ (Fig. 5e, f) perovskite at 4 K temperature. There are four exciton bands in both the perovskites, with peak absorption of the first three bands at 410, 440 and 480 nm. The FE has an absorption peak at 480 nm and an emission peak at 490 nm (shown by red dashed lines in Fig. 5c, e). The emission of the FE could only be collected with excitation up to 440 nm due to the presence of a 450 nm edge pass filter, leaving a blank region in the top left region of the colourmap (Fig. 5c, e). Emission from the two higher exciton states was not observed, perhaps due to ultrafast relaxation into the FE exciton. While red emission from the broadband was observed with all excitation wavelengths from 400 to 500 nm, the maximum intensities were obtained when excited directly into the three higher exciton bands. From the PLE spectra of $[H_3N(CH_2)_8NH_3]PbI_4$ (Fig. 5d), it is evident that the broadband is populated by relaxation from the FE (also blue dashed line in Fig. 5c). This shows that the broadband emission in the parent perovskite does not arise from a permanent defect state and instead forms an intrinsic STE state. On the other hand, while the intercalated perovskite largely retains its original band structure of the three higher excitonic states, the PLE intensities are altered relative to one another compared to the parent perovskite (Fig. 5f). While a clear absorption peak of the broadband emission at energies lower than the FE is not observed, there is a non-negligible absorption below the FE. From this result, we can infer that intercalation modifies the nature of the STE due to modification of the band structure. This result is also supported by computational studies.

To exclude extrinsic factors, no broadband emissions or any pronounced edge emission were observed in combined microscopy and PL experiments of $[H_3N(CH_2)_8NH_3]PbI_4$ and $[H_3N(CH_2)_8NH_3]PbI_4 \cdot I_2$, for single crystals or exfoliated flakes at room temperature (Supplementary Figs. 31, 32). Interstitial iodide defects acting as colour centres have been shown to provide occupied in-gap states in the electronic structure, which enhances white-light emission[44]. However, in the intercalated perovskites, the spaces between adjacent $[PbX_4]_\infty$ layers are occupied by iodine molecules and $[H_3N(CH_2)_mNH_3]^{2+}$ cations, which will change the defect formation energy and therefore the resulting defect concentrations. Layered bromide perovskites which show octahedral distortions or perovskites which are derived from slicing in the (110) direction often exhibit broadband emissions with large Stoke shifts which have been linked to STEs[45–47]. Only a few lead iodide-based layered perovskites show weak, broad emissions at low temperature. As only small changes in octahedral distortion were observed upon $I_2$ intercalation in $[H_3N(CH_2)_8NH_3]PbI_4$, we think that conformational changes of $[H_3N(CH_2)_mNH_3]^{2+}$cations caused by iodine intercalation, which yields a different effective cation radius[48–51], may result in a different rotational and vibrational mode of $[H_3N(CH_2)_8NH_3]^{2+}$. A different phonon formation is expected for $[H_3N(CH_2)_8NH_3]PbI_4 \cdot I_2$ when compared to $[H_3N(CH_2)_8NH_3]PbI_4$, as inorganic layers have a strong vibrational coupling with the intercalated iodine in order to maintain halogen bonds. Computational studies have shown that the phonon that couples to the exciton could reside on the $[H_3N(CH_2)_mNH_3]^{2+}$ cation, rather than on the inorganic layer[52]. Based on the experimental results, broadband emission of the intercalated perovskites may result from a variety of factors. A precise estimation of the trap energy requires a temperature-dependent measurement of both the emission lifetimes and photoluminescence quantum yield, that is part of our next study.

Preliminary photovoltaic characterisation has been carried out on $[H_3N(CH_2)_8NH_3]PbI_4$ and $[H_3N(CH_2)_8NH_3]PbI_4 \cdot I_2$ (Supplementary Figs. 33–36). Although these devices require much more optimisation in terms of factors such as film thickness and morphology, the measurements show that these materials have potentially useful photovoltaic properties. Based on the preliminary device testing, the intercalated materials show higher short circuit current densities than their parent layered perovskites, as expected, but the reduction in band gap has a bigger overall effect on device performance, as indicated by the smaller open circuit voltage in the intercalated materials.

In summary, our work integrates first-principles calculations and experimental work in the developing field of intercalation in hybrid perovskites. Under the guidance of theoretical calculations, six intercalated perovskites have been successfully synthesised, characterised by single-crystal XRD and their optical properties measured. By carrying out a comprehensive, systematic study, we have developed a series of structural guidelines which include conformational flexibility in the $[H_3N(CH_2)_mNH_3]^{2+}$cation, halogen bond length, $X$-$X$-$X$ bond angles (between halogen ions/atoms) and halide-halide distance that can be used to determine whether intercalation in hybrid perovskites is possible. Size mismatch between the halogen atoms in the intercalated molecule and the halide ions in the $[PbX_4]_\infty$ layers, which contain smaller halides, induces an expansion of the inorganic layers, distorts the octahedra and leads to promising band gaps. On the basis of these conclusions, the intercalated perovskite $[H_3N(CH_2)_6NH_3]PbI_4 \cdot Br_2$ was predicted to have a promising band gap of 1.51 eV, which is close to that of 3D hybrid perovskites, such as $CH_3NH_3PbI_3$.

Highly oriented thin films of intercalated perovskites, including $[H_3N(CH_2)_6NH_3]PbI_{4-y}Br_y \cdot xBr_2$, were prepared through a post-synthesis intercalation method. Photoluminescence studies of the intercalated perovskite samples show that intercalation allows tuneability of the STE energy. Intercalation also modifies the nature of the STE to permit direct, albeit weak, excitation into the intercalated band. We expect intercalation to be applicable to the layered hybrid perovskites containing metals other than Pb, such as Sn and Ge. In addition, it opens up the possibility for exploring 'host–guest' chemistry in organic-inorganic metal halides.

## Methods
### DFT calculations
DFT calculations were performed using the Vienna ab initio simulation package (VASP)[53]. The projected augmented wave (PAW)[54,55] method was used to account for the effect of core electrons on valence electron density. The geometries of the most stable configurations for host layered hybrid perovskites, guest molecules and intercalated structures were obtained through energy minimisation by PBE

functional with DFT-D3 vDW correction[56]. A $2 \times 4 \times 4$ k-point grid and 550 eV plane-wave cutoff were used for the optimisation of hosts and intercalated structures after converging testing, and all forces acting on the ions were below 0.01 eV/Å after relaxation. All reported host structures were from the 2D perovskite database[57], while other theoretical structures were generated by replacing the halides and $[H_3N(CH_2)_mNH_3]^{2+}$ cations (where $m = 5–10$) in reported structures, which was followed by cell optimisation. The band structure calculations were performed with HSE06 hybrid functional[58]. To balance the efficiency and accuracy of calculation, we didn't include the spin-orbit coupling (SOC) because it shows limited influence on the band gap value of $[H_3N(CH_2)_6NH_3]PbI_4 \cdot Br_2$ (1.91 eV with SOC[15] and 1.98 eV without SOC in this research).

## Synthesis and characterisation of $[H_3N(CH_2)_mNH_3]PbX_4$ and $[H_3N(CH_2)_mNH_3]PbX_4 \cdot X_2$

**Starting materials.** 1,7-Diaminoheptane ($H_2N(CH_2)_7NH_2$, ≥98%), 1,8-diaminooctane ($H_2N(CH_2)_8NH_2$, ≥98%), 1,9-diaminononane ($H_2N(CH_2)_9NH_2$, ≥99%), lead (II) iodide ($PbI_2$, ≥98%), lead (II) bromide ($PbBr_2$, ≥98%), bromine ($Br_2$, 99.8%), iodine ($I_2$, 99.5%), iodine monobromide (IBr, 98%), hydriodic acid (HI,57% w/w aq. soln., stabilised with 1.5% hypophosphorous acid) and hydrobromic acid (HBr, 48%, w/w aqueous solution) were purchased from Alfa Aesar. All chemicals were directly used without further purification.

## Preparation of intercalated crystalline samples

**Single-crystal growth.** In order to obtain single crystals of the intercalated perovskites, all of the reagents were placed in a sealed 30 mL Teflon-lined stainless-steel autoclave, which was then placed into an oven firstly at temperatures in the region of 120 to 160 °C (see Table S2) and secondly at 80 °C. The resulting crystals were filtered and naturally dried at room temperature.

**Preparation of $[H_3N(CH_2)_mNH_3]PbX_4$ parent samples.** The parent, layered hybrid perovskites, $[H_3N(CH_2)_mNH_3]PbX_4$, with $m = 8$ and 9 were be prepared as single crystals as previously reported[59]:

**$[H_3N(CH_2)_8NH_3]PbBr_4$.** $PbBr_2$ (0.734 g, 2 mmol) was dissolved in concentrated HBr (8 mL) with moderate heating and stirring. Once the $PbBr_2$ had dissolved, $H_2N(CH_2)_8NH_2$ (0.292 g, 2 mmol) was added to the warm mixture. The temperature of this mixture was increased to 90 °C, with vigorous stirring, until all precipitates disappeared. The resulting colourless/pale-yellow solution was left to stand at 50 °C for 24 h, so that most of the product would form as colourless chip-shaped crystals of $[H_3N(CH_2)_8NH_3]PbBr_4$. The product was filtered and dried in an oven at 60 °C for 12 h.

**$[H_3N(CH_2)_8NH_3]PbI_4$.** $PbI_2$ (0.922 g, 2 mmol) was dissolved in concentrated HI (10 mL) with moderate heating and stirring. Once the $PbI_2$ had dissolved, $H_2N(CH_2)_8NH_2$ (0.292 g, 2 mmol) was added to the warm mixture. The temperature of this mixture was increased to 90 °C, with vigorous stirring, until all precipitates disappeared. The resulting pale-yellow solution was left to stand at 50 °C for 24 h, so that most of the product would form as yellow chip-shaped crystals of $[H_3N(CH_2)_8NH_3]PbI_4$. The product was filtered and dried in an oven at 60 °C for 12 h.

**$[H_3N(CH_2)_9NH_3]PbI_4$.** $PbI_2$ (0.922 g, 2 mmol) was dissolved in concentrated HI (10 mL) with moderate heating and stirring. Once the $PbI_2$ had dissolved, $H_2N(CH_2)_9NH_2$ (0.321 g, 2 mmol) was added to the warm mixture. The temperature of this mixture was increased to 90 °C, with vigorous stirring, until all precipitates disappeared. The resulting pale-yellow solution was left to stand at 50 °C for 24 h, so that most of the product would form as yellow chip-shaped crystals $[H_3N(CH_2)_9NH_3]PbI_4$. The product was filtered and dried in an oven at 60 °C for 12 h.

**Preparation of Intercalated $[H_3N(CH_2)_mNH_3]PbX_4 \cdot X_2$ polycrystalline samples.** Grinding single crystals of the intercalated samples into a polycrystalline form may de-intercalate the halogen molecules[15]. Therefore, a two-step intercalation method (combining both solution and solid-state methods) was developed to synthesise four polycrystalline, $I_2$-intercalated materials.

**$[H_3N(CH_2)_8NH_3]PbBr_4 \cdot I_2$.** Solid $[H_3N(CH_2)_8NH_3]PbBr_4$ and solid $I_2$ were mixed in 1:1 molar ratio and ground at room temperature, in a pestle and mortar for 10 min. PXRD showed that at room temperature, the resulting dark-yellow powder contained $[H_3N(CH_2)_8NH_3]PbBr_4 \cdot I_2$ as the major phase and a $C$-centred monoclinic phase as a secondary phase. Due to equipment availability, we have been unable to get low-temperature PXRD data to match the data collection temperature of the SCXRD data.

**$[H_3N(CH_2)_8NH_3]PbI_4 \cdot I_2$.** Solid $[H_3N(CH_2)_8NH_3]PbI_4$ and solid $I_2$ were mixed in 1:1 molar ratio and ground at room temperature, in a pestle and mortar for 10 min. PXRD showed that the resulting dark-red powder was pure $[H_3N(CH_2)_8NH_3]PbI_4 \cdot I_2$.

**$[H_3N(CH_2)_9NH_3]PbI_4 \cdot I_2$.** Solid $[H_3N(CH_2)_9NH_3]PbI_4$ and solid $I_2$ were mixed in 1:1 molar ratio and ground at room temperature, in a pestle and mortar for 10 min. PXRD showed that the resulting dark-red powder was pure $[H_3N(CH_2)_9NH_3]PbI_4 \cdot I_2$.

**$[H_3N(CH_2)_7NH_3]PbI_4 \cdot I_2$.** In the exploration of the synthesis of $[H_3N(CH_2)_7NH_3]PbI_4 \cdot I_2$ we found that preferred product from our standard synthetic method was $[(H_3N(CH_2)_7NH_3)_4Pb_3I_{12} \cdot 2I^-]$, which has already been crystallographically characterised by others[60]. Therefore we attempted the synthesis of the desired $[H_3N(CH_2)_7NH_3]PbI_4 \cdot I_2$ by first preparing an intermediate compound, $[H_3N(CH_2)_7NH_3]I_2 \cdot I_2$. $H_2N(CH_2)_7NH_2$ (0.321 g, 2 mmol) and $I_2$ (0.508 g, 2 mmol) was dissolved in concentrated HI (10 mL) with moderate heating and stirring. Once the solution was clear, it was cooled to room temperature slowly to obtain the dark-yellow crystalline $[H_3N(CH_2)_7NH_3]I_2 \cdot I_2$[61].

Solid $[H_3N(CH_2)_7NH_3]I_2 \cdot I_2$ and solid $PbI_2$ were mixed in a 1:1 molar ratio and ground manually at room temperature, in a pestle and mortar for 10 min. Then they were pressed into a pellet under 1 ton pressure and heated at 160 °C for 60 min. PXRD showed that the resulting material is $[H_3N(CH_2)_7NH_3]PbI_4 \cdot I_2$.

## Characterisation

**X-ray diffraction.** Single-crystal X-ray diffraction data were collected at low temperature (173 K) or room temperature (298 K) on either a Rigaku FR-X Ultrahigh Brilliance Microfocus RA generator/confocal optics, or a Rigaku SCX Mini diffractometer, using Mo−Kα radiation. Data were collected using *CrystalClear* (Rigaku) software[62]. Absorption corrections were performed empirically from equivalent reflections based on multiscans using either *CrystalClear*[62] or *CrysAlisPro*[63]. Structures were solved by direct methods using *SHELXT*[64], and full-matrix least-squares refinements on $F^2$ were carried out using *SHELXL-2019/2*[65] incorporated in the WinGX programme[66]. Non-H atoms were refined anisotropically, and hydrogen atoms were treated as riding atoms. Restraints on C−C and C−N bond lengths were applied. Further data collection at ambient temperature was attempted on crystals of $[H_3N(CH_2)_8NH_3]PbBr_4 \cdot I_2$. However, crystal quality problems meant that a dataset could not be collected at room temperature, which would allow full structure refinement. The best unit cell parameters were obtained in space group $C2$, with $a = 8.26$ Å, $b = 8.28$ Å, $c = 30.40$ Å and $\beta = 92.25$ Å, which showed the perovskite framework of the structure, but the organic ammonium cation was disordered and the C/N positions could not be reliably determined. These unit cell parameters were, however, suitable to be used, in combination with those of the 173 K structure of $[H_3N(CH_2)_8NH_3]$

$PbBr_4 \cdot I_2$, in the Pawley fit to the PXRD data for this structure (Supplementary Fig. 10).

Ambient temperature powder X-ray diffraction data were collected on a PANalytical Empyrean diffractometer, equipped with an X'Celerator detector, using Cu $K_{\alpha 1}$ ($\lambda = 1.5406$ Å) radiation in the range of $2\theta = 3 - 40°$, with a step size of $0.017°$ and a time per step of 0.913 s.

**UV-visible spectroscopy.** Diffuse reflectance UV−visible spectra were collected on polycrystalline powders of all samples, using a JASCO-V650 ultraviolet−visible spectrophotometer with a wavelength range of 190 − 900 nm. $BaSO_4$ was used as a reference.

**Raman spectroscopy.** Raman spectroscopy was carried out on a Renishaw in-Via Qontor microscope, using a 532 nm laser.

**Thermal gravimetric analysis (TGA).** TGA experiments were conducted in air on a Netzsch STA 449C equipped with a mass spectrometer using a heating rate of 5 °C min$^{-1}$ in the temperature range 25–250 °C.

**Photoluminescence (PL).** The samples were cooled using an Oxford Instruments MicrostatHe liquid helium flow cryostat. Light from a 415 nm constant wavelength laser (Qioptik iFlex-2000) was used to excite the films, and photoluminescence spectra were collected using an Andor Shamrock-750 spectrograph coupled to an Andor Newton EMCCD.

For photoluminescence excitation spectra, a NKT Supercontinuum laser with tunable pulsed excitation at 80 MHz repetition rate, was used to excite the sample at 4 K. A 450 nm (550 nm) short-pass filter was used to clean the laser and another 450 nm (550 nm) long pass filter was used in the detection arm to block the laser while collecting emission from the narrow band (broadband).

**Preparation of samples for photoluminescence.** Polymethyl methacrylate (PMMA) colloidal solution: 40 mg PMMA is dissolved into 1 mL of chlorobenzene with 3 min moderate stirring and 80 °C heating to form a colloidal solution.

Thin film samples: Perovskite thin films were spin-coated (see below) onto sapphire glass substrates. Then, one drop of the PMMA colloidal solution was added on top of the perovskite thin films and spun at 3000 rpm for 65 s. The substrate was annealed at 80 °C for 60 s. The PMMA coverage can prevent halogen release under vacuum.

Single-crystal samples: Crystalline perovskite samples were placed onto sapphire glass substrates. Then, one drop of the PMMA colloidal solution was added on top of the single crystals and spun at 3000 rpm for 65 s. The substrate was annealed at 80 °C for 60 s. The PMMA coverage can prevent halogen release under vacuum and helps hold the crystals in place on the substrate.

**Optical microscopy under illumination.** Crystalline perovskites were placed on microscope slides, illuminated with wavelengths selected from a mercury lamp (425, 515 and 590 nm), magnified in a Leica DMIRE2 microscope, and imaged with a Hamamatsu C4742 CCD camera.

**Combined optical microscopy and photoluminescence.** $[H_3N(CH_2)_8NH_3]PbI_4 \cdot I_2$ crystals were also put under an optical microscope (Leica DMIRE2 microscope connected to the spectrometer with a mercury lamp and imaged with a Hamamatsu C4742 CCD camera), so that microscopy images could be obtained at the same time as PL experiments. This allows us to probe the potential influence on the broad emission caused by the thickness of the sample.

**Thin film uniformity studies using optical microscopy.** The optical microscopy images of the crystalline films were recorded using a Nikon Eclipse LV1000D microscope.

## Thin film fabrication and morphology
### Film fabrication
**Substrate cleaning.** Quartz-coated glass substrates (purchased from Ossila) were cleaned with distilled water, isopropanol, and acetone for 20 min each in an ultrasonic bath. The clean substrates were heated at 80 °C on a hotplate and dried using a flow of compressed Argon for 5 min. Then the substrates were plasma-ashed (with a Mini Flecto 1320010 manufactured by Gala Instrumente GmbH) using UV ozone for 3 min.

### $[H_3N(CH_2)_8NH_3]PbI_4 \cdot I_2$ films
**Iodine-vapour method for (002)-oriented films**
**Solution preparation.** A 1 M solution was prepared by dissolving polycrystalline $[H_3N(CH_2)_8NH_3]PbI_4$ in 1 mL of anhydrous DMF. Then, 0.1 mL of HI was added into the solution. The solution was then heated moderately and stirred until all the precipitates were dissolved.

**Spin coating.** Three drops of 1 M $[H_3N(CH_2)_8NH_3]PbI_4$ solution were placed on the quartz-coated side of the substrate, and the substrate was spun at 3000 rpm for 40 s to fabricate films. The substrate was annealed at 120 °C for 2 min.

**Intercalation of films.** About 2 g of solid $I_2$ was placed in a closed flask and heated at 150 °C, until all iodine is in vapour form. The $[H_3N(CH_2)_8NH_3]PbI_4$ film was then dipped into the iodine vapour for 5 s.

**Iodine-vapour method for randomly oriented films**
**Spin coating.** Three drops of a 1 M $[H_3N(CH_2)_8NH_3]PbI_4$ solution (in anhydrous DMF) were placed on the quartz-coated side of the substrate, and the substrate was spun at 3000 rpm for 40 s to fabricate films.

**Intercalation of films.** About 2 g of solid $I_2$ was placed in a closed flask and heated at 150 °C, until all iodine is in vapour form. The $[H_3N(CH_2)_8NH_3]PbI_4$ film was dipped into the iodine vapour for 5 s immediately after the spin coating.

**Direct method for randomly oriented films**
**Solution preparation.** A 1 M solution was prepared by dissolving the single crystalline/polycrystalline $[H_3N(CH_2)_8NH_3]PbI_4 \cdot I_2$ in 1 mL of anhydrous DMF. Then, 0.1 mL of HI was added into the solution. The solution was then heated moderately and stirred until all precipitates dissolved.

**Spin coating.** Three drops of a 1 M $[H_3N(CH_2)_8NH_3]PbI_4 \cdot I_2$ solution were placed on the quartz-coated side of the substrate and the substrate was spun at 2500 rpm for 65 s to fabricate films. The substrate was annealed at 80 °C for 30 s.

### $[H_3N(CH_2)_6NH_3]PbI_4 \cdot Br_2$ films
**Bromine-vapour method for (002)-oriented film**
**Solution preparation.** A 1 M solution was prepared by dissolving the polycrystalline $[H_3N(CH_2)_6NH_3]PbI_4$ in 1 mL of anhydrous DMF. Then, 0.1 mL of HI was added into the solution and following by moderate heating and stirring until all precipitates are dissolved.

**Spin coating.** Three drops of 1 M $[H_3N(CH_2)_6NH_3]PbI_4$ solution were placed on the quartz-coated side of the substrate, and the substrate was spun at 3000 rpm speed for 40 s to fabricate films. The substrate was annealed at 120 °C for 2 min.

**Intercalation of films.** About 2 g of liquid $Br_2$ was placed in a closed flask, which was cooled in a dry ice bath. The $[H_3N(CH_2)_6NH_3]PbI_4$ film was held 3 cm above the bromine liquid level for 5 s.

## Photovoltaic device fabrication

To investigate the potential application of the intercalated perovskites in photovoltaics, a typical n-i-p perovskite solar cell device (ITO/SnO$_2$/ intercalated perovskite/Spiro-OMeTAD/Au) was fabricated as shown in Supplementary Fig. 33.

1 sun $J$–$V$ measurements were carried out by using a solar simulator with a xenon arc lamp (150 W, 50 × 50 mm, Class AAA, Sciencetech Solar simulator) at the irradiance level of 100 mW cm$^{-2}$ (AM 1.5 G). The measurement active area of the devices was defined by a customised aperture mask of 0.05 cm$^2$. The current–voltage measurement settings were selected as follows: voltage settling time: 0.2 s; voltage increment: 0.05 V; scan rate: 0.2 V s$^{-1}$. The resulting J–V curves are given in Supplementary Figs. 35, S36.

## Data availability

The research data underpinning this publication can be accessed at https://doi.org/10.17630/4067ee08-80db-4be7-8d34-7504f86a8dbd.

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

## Acknowledgements

J.L.P. thanks the University of St Andrews for funding and the Carnegie Trust for a Research Incentive Grant (RIG008653). J.L.P. and H.O. thank the Leverhulme Trust for a Research Grant (RPG-2022-188). We also thank EPSRC for funding (EP/T019298/1, EP/R023751/1 and EP/V034138/1). L.K.J. thanks UKRI for a Future Leaders Fellowship (MR/T022094/1). This work used the ARCHER2 UK National Supercomputing Service (https://www.archer2.ac.uk) and high-performance computing resources at the University of Liverpool (M.S.D.). H.-Y.T.C. thanks the National Science and Technology Council (NSTC) in Taiwan (111-2221-E-007-087-MY3, 111-2112-M-007-028-MY3 and 112-2113-M-007-004) and National Tsing Hua University (112Q2711E1 and 112QI014E1).

## Author contributions

L.-J.Y. prepared samples, collected data, analysed data and fabricated films. W.X., H.-Y.T.C. and M.S.D. carried out DFT calculations. L.-J.Y., S.H., S.K.R. and H.O. collected and analysed PL data. S.W., L.K.J. did device measurements. D.B.C. and A.M.Z.S. collected SC-XRD data. D.N.M., L.-J.Y. and J.L.P. collected Raman data. L.-J.Y., D.B.C. and J.L.P. analysed XRD data. L.-J.Y., W.X., M.S.D. and J.L.P. wrote the manuscript, with contributions from all co-authors. L.-J.Y., W.X., M.S.D. and J.L.P. conceived and coordinated the project.

## Competing interests

The authors declare no competing interests.
