## [Transparent Peer Review file · Nature Communications]

Advancing Intercalation Strategies in Layered Hybrid Perovskites by Bringing Together Synthesis and Simulations

Corresponding Author: Dr Julia Payne

Version 0:

Reviewer comments:

Reviewer #1

(Remarks to the Author)

Manuscript (ID: NCOMMS-24-21839-T) reports discoveries of six halogen intercalated $[\text{H}_3\text{N}(\text{CH}_2)_m\text{NH}_3]\text{PbX}_4\cdot\text{X}_2$ ($m = 7, 8, 9$; X = halide) new layered lead-halide perovskites. Broadly, the work includes structure optimization, calculations of electronic structures, and materials synthesis and X-ray structures. The authors earlier reported one such compound $[\text{H}_3\text{N}(\text{CH}_2)_6\text{NH}_3]\text{PbBr}_4\cdot\text{Br}_2$ (ref. 7) and in this submission they have provided generality of their approach over several systems of similar kind. What is exciting about these materials is that they retain structural relationship to the parent lead-halide perovskites with adjusted electronic structures due to halogen intercalation, resulting in significant reduction in the optical bandgaps. The bandgaps of some of these compounds demonstrate a range of potential optoelectronic applications. While the work is exciting, and the design strategy is sound that will likely warrant several following additions, the work is not present well. To position this work more clearly in the literature, I recommend following changes before accepting for publication.

In the introduction, authors have included only one paper as their motivation to carry out intercalation chemistry on layered halide perovskites which is their own paper (ref. 7). There are other studies known in this area for different applications. For example, DMSO intercalation in $(\text{OHC}_6\text{H}_5\text{CH}_2\text{CH}_2\text{NH}_3)_2\text{PbBr}_4$ (Adv. Mater. Technol. 2023, 8 (5), 2201465). Authors should provide broader view of the area in the introduction.

As per the theoretical calculations carried out here, the $[\text{H}_3\text{N}(\text{CH}_2)_m\text{NH}_3]\text{PbX}_4\cdot\text{X}_2$ ($m = 7$) compounds are among the most stable ones with high negative E_{binding} energies. The most stable compound among all the intercalated compounds studied in this submission is $[\text{H}_3\text{N}(\text{CH}_2)_7\text{NH}_3]\text{PbI}_4\cdot\text{Br}_2$ with the E_{binding} of -0.62 eV. However, the authors could not isolate this compound as single crystal or as thin film (Table S8), but still they have projected this as the most promising photovoltaic materials among all the compounds on page number 5. In this case the theoretical results and experimental observations contradict each other. Not only this, none of the other proposed compounds with $m = 7$ were accessible as thin films but were isolated as single crystals. Why are they unable to make their most stable compound?

“A total of 54 compounds were screened, with eleven compositions predicted to exist, having E_{binding} less than -0.16 eV.” What is so special about -0.16 eV?

Authors have written “excellent agreement between the experimental PXRD pattern collected at room temperature” however I find that the matching of experimental and simulated patterns is poor in some of the latter members of $[\text{H}_3\text{N}(\text{CH}_2)_m\text{NH}_3]\text{PbX}_4\cdot\text{X}_2$ family. Figure S10- $[\text{H}_3\text{N}(\text{CH}_2)_8\text{NH}_3]\text{PbBr}_4\cdot\text{I}_2$: Position mismatch for the peaks at $2\theta = \sim 15^\circ, \sim 22^\circ \sim 29^\circ, \sim 33^\circ$. Here, I have mentioned only the tall peaks, but the mismatch is there in almost all the higher 2θ peaks. Similarly Figure S11, $[\text{H}_3\text{N}(\text{CH}_2)_8\text{NH}_3]\text{PbI}_4\cdot\text{I}_2$, the peaks at $2\theta = \sim 15^\circ, \sim 22^\circ \sim 29^\circ$ are mismatching with the simulated pattern. How can authors say that these are excellent matches? Author should redo these measurement or explain the differences.

Optical absorption spectra are confusing. Authors have written...“Of the bulk samples synthesised, $[\text{H}_3\text{N}(\text{CH}_2)_7\text{NH}_3]\text{PbI}_4\cdot\text{I}_2$ had the lowest band gap (1.74 eV)”, which does not seem correct to me. From the UV spectra (Figure S17), it is clear that the bandgaps of $[\text{H}_3\text{N}(\text{CH}_2)_m\text{NH}_3]\text{PbI}_4\cdot\text{I}_2$, are in the order $m = 9 < m = 7 < m = 8$. But authors have listed in Table S7 in the sequence of $m = 7 < m = 8 < m = 9$. How were the bandgaps determined? They should reassess the bandgap values.

Discussion on the photophysical properties needs to be strengthened as most of the information provided is about their “observations” rather than comparison to the established literature.

Page 13, lines 17 and 18: the sequence ($m = 8 < m = 7 < m = 9$) of red emission spectra is provided in “wavelength” unit, while that of the absorption spectra in “Energy” unit ($m = 7 < m = 8 < m = 9$). For easy comparison, the sequences need to be in the same measurement unit (i.e. “wavelength” or “energy”). Similarly, for the sequence provided on line 8 of this page about the green emission $m = 8 < m = 7 < m = 9$).

Why are the measurements carried out at particularly 4 K?

How did authors confirm that in $[\text{H}_3\text{N}(\text{CH}_2)_7\text{NH}_3]\text{PbBr}_4\cdot\text{IBr}$, the intercalant has I and Br in 1:1 ratio rather than an alloyed variant of IBr with more iodine content than bromine. The I-Br bond distance in the CIF is close to that of the I-I distance of I_2 molecules and far from the Br-Br distance of Br_2 molecule.

Reviewer #2

(Remarks to the Author)

The manuscript by Payne and coworkers discusses the intercalation of neutral, molecular X_2 species into 2D hybrid halide perovskites as a means of tuning the electronic structure. The manuscript requires major revisions:

1. The title is long and the first clause is somewhat pointless.
1. The authors do a poor job of introducing the subject of intercalation of molecules into 2D perovskites, including molecular dihalogens. They do a better job of this in their previous publication on the topic (reference 7) but the prior work of others is ignored here completely.
2. The layered perovskites studied here are clearly of the Dion-Jacobson type, but the introduction only discusses R-P phases.
3. Nowhere in the manuscript is a complete structure displayed that shows the 2D perovskite slabs with the amines and the X_2 intercalant. This would be helpful to the reader, and one of the figures from the SI could be moved.
4. The authors may have missed the importance of even/odd alteration of the α - ω diamines. Odd chains pack poorly giving rise to less stable hybrid halides. They can potentially be more readily stabilized by X_2 intercalation, but also, the poorer chain packing may help admit X_2 .
5. The trend in the PL also reflects this.
6. What is the excitation wavelength for the PL. The PL has a broad band emission that is attributed to the intercalant. Can they excite into the band directly? PL excitation spectra may be informative. The explanation of the breadth is incomprehensible. Please note that a molecular species that is not held strongly, is very likely to have the phonon landscape that would lead to broad emission. On this point, STEs are referred to as being extrinsic to the material. They are not.
7. In the conclusion, the band-band emission is due to STEs but this is in seeming contradiction with the previous discussion.
8. On the whole, the quality of figures is poor. There is extensive use of bar graphs that make no sense and these are difficult to read. These are all cases where scatter plots make more sense. Also, figures with lots of too-small text are never appealing.

Reviewer #3

(Remarks to the Author)

Version 1:

Reviewer comments:

Reviewer #1

(Remarks to the Author)

Authors have addressed my comments, accept a few of them as given below-

Please add the method that was used for calculating the layer-shift factors (Ls).

On page 4, the authors write, “In addition, we note that the single crystal structures published had high R-factors and large

residual electron densities, which may indicate issues with the crystal structure or refinement.(ref. 14) (PEA-OH)PbBr₄·DMSO also could be used as a photodetector.” I do not agree with this statement as the single-crystal X-ray data and the PXRD data reported in this ref. 14 are of better quality than some of the compounds reported in this submission (Table S3). For example, the R-factor of [H₃N(CH₂)₇NH₃]PbBr₄·IBr is 12.5% and has high electron density, which is not modeled. R-factors of [H₃N(CH₂)₈NH₃]PbI₄·I₂ and [H₃N(CH₂)₉NH₃]PbI₄·I₂ are also high. So the authors should remove/reword this statement.

I do agree with the authors that the unit cell is expected to increase in size as temperature is raised and it is obvious in most of the Rietveld refined PXRD data that they have provided in the revised work. However, I do not fully agree with the author's explanation about the mismatch of some of the PXRD data. For example, in the case of [H₃N(CH₂)₈NH₃]PbBr₄·I₂ (Figure S10), there are two intense peaks at 2θ ~ 15 ° and ~ 23 ° angles, which are absent in the simulated pattern. These peaks could be due to some impurity/secondary phase but can not be due to the difference in temperature of single crystal and powder diffraction data. Similarly, in [H₃N(CH₂)₇NH₃]PbBr₄·IBr (Figure S8), there is an extra peak at ~ 2θ ~ 7 °. So, some discussion on PXRD data needs to be added to the manuscript to indicate the mismatch. Also, please clarify why were the single crystal diffraction data collected in some cases at 173 K and others at 298 K.

Reviewer #2

(Remarks to the Author)

I am happy with the changes. The work can be published, in my opinion.

Reviewer #3

(Remarks to the Author)

Version 2:

Reviewer comments:

Reviewer #1

(Remarks to the Author)

The authors have addressed my comments and, in my opinion, the comments of other referees. The article can be accepted for publication in its current form.

Open Access This Peer Review File is licensed under a Creative Commons Attribution 4.0 International License, which permits use, sharing, adaptation, distribution and reproduction in any medium or format, as long as you give appropriate credit to the original author(s) and the source, provide a link to the Creative Commons license, and indicate if changes were

made.

REVIEWER COMMENTS

Reviewer #1 (Remarks to the Author):

Manuscript (ID: NCOMMS-24-21839-T) reports discoveries of six halogen intercalated $[H_3N(CH_2)_mNH_3]PbX_4 \cdot X_2$ ($m = 7, 8, 9$; $X =$ halide) new layered lead-halide perovskites. Broadly, the work includes structure optimization, calculations of electronic structures, and materials synthesis and X-ray structures. The authors earlier reported one such compound $[H_3N(CH_2)_6NH_3]PbBr_4 \cdot Br_2$ (ref. 7) and in this submission they have provided generality of their approach over several systems of similar kind. What is exciting about these materials is that they retain structural relationship to the parent lead-halide perovskites with adjusted electronic structures due to halogen intercalation, resulting in significant reduction in the optical bandgaps. The bandgaps of some of these compounds demonstrate a range of potential optoelectronic applications. While the work is exciting, and the design strategy is sound that will likely warrant several following additions, the work is not present well. To position this work more clearly in the literature, I recommend following changes before accepting for publication.

We thank the reviewer for this positive response. We will address your suggestions for improvements in the text below.

In the introduction, authors have included only one paper as their motivation to carry out intercalation chemistry on layered halide perovskites which is their own paper (ref. 7). There are other studies known in this area for different applications. For example, DMSO intercalation in $(OHC_6H_5CH_2CH_2NH_3)_2PbBr_4$ (*Adv. Mater. Technol.* 2023, 8 (5), 2201465). Authors should provide broader view of the area in the introduction.

We apologise for lack of background on intercalation in layered halide perovskites. Of course, we should cite the appropriate papers and we have modified the introduction accordingly. We thank the reviewer for pointing out the *Adv. Mater. Technol.* 2023, 8, (5), 2201465 paper and we have included it in the references (Ref 14 in the main article). The new text has been added to the introduction and reads as follows:

In 1986, Maruyama *et al.* reported that small molecules, including 1-chloronaphthalene, *o*-dichlorobenzene and hexane, could be reversibly intercalated into layered hybrid perovskites $(C_{10}H_{21}NH_3)_2CdCl_4$ and $(C_9H_{19}NH_3)_2PbI_4$.¹ To the best of our knowledge, this was the first report of intercalation in layered hybrid perovskites. However, in this study, single crystals of the intercalated compound were not obtained and only changes in unit cell parameters could be observed.¹ Mitzi *et al.* then looked at the intercalation of C_6H_6 and C_6F_6 into $(C_6F_5C_2H_4NH_3)_2SnI_4$ and $(C_6H_5C_2H_4NH_3)_2SnI_4$ respectively.² In this instance, intercalation of C_6F_6 into $(C_6H_5C_2H_4NH_3)_2SnI_4$ only resulted in an 0.04 eV change in band gap, despite the distance between the $[SnI_4]_{\infty}$ layers changing from 16.3 Å to 20.6 Å.² More recently, intercalation has played an important role in the processing of organic-inorganic metal

halides, as solvents such as DMF etc have been postulated to intercalate between PbI_2 layers.³⁻⁵ Nag has also looked at intercalation in a number of compounds, including $(\text{BA})_2\text{PbI}_4$ (where BA = butylammonium), and $(\text{PEA})_2\text{PbI}_4$ (where PEA = phenylethylammonium), but we note that no single crystal structures were obtained from single crystal X-ray diffraction.⁶ In this work, $(\text{BA})_2\text{PbI}_4$ displayed two peaks in the photoluminescence spectrum, which was attributed to two different areas of the crystal (edge and terrace), which suggested electronic interactions between neighbouring $[\text{PbI}_4]_\infty$ layers.⁶ When iodine was intercalated, only a single emission was observed in the photoluminescence and this was found at higher energies.⁶ In this study, the lower energy peak had been attributed to edge emission. This process was reversible. The same group then went to look at hexane intercalation into $(\text{DA})_2\text{PbI}_4$ (where DA = decyl ammonium), which again changed the PL emission from dual to single emission.⁶ However, the intercalated molecules were prone to deintercalation, which precluded the growth of crystals suitable for single crystal X-ray diffraction studies.⁶ As a result, $(\text{PEA})_2\text{SnI}_4 \cdot \text{C}_6\text{F}_6$, previously prepared by Mitzi *et al.* was investigated.^{2,6} It also showed dual emission in the PL spectra and like the other compounds, the low energy PL emission disappeared upon intercalation of the C_6F_6 molecule.⁶ To complete the study, Nag *et al.* also looked at intercalation in $(\text{C}_m\text{H}_{2m+1}\text{NH}_3)_2\text{PbI}_4$, where the length of the carbon chain was systematically varied.⁶ As the carbon chain length increased, the PL went from dual emission to single emission, with the loss of the low energy peak.⁶ Karunadasa looked at the intercalation of I_2 into $(\text{CH}_3(\text{CH}_2)_5\text{NH}_3)_2\text{PbI}_4$ and the related compound containing a terminal alkyl iodide group, $(\text{ICH}_2(\text{CH}_2)_5\text{NH}_3)_2\text{PbI}_4$.⁷ In these compounds, I_2 was only stable for a short time and no single crystal XRD could be obtained for either material, preventing full structural characterisation of these materials.⁷ We note that the intercalation of I_2 was found to be more stable in $(\text{ICH}_2(\text{CH}_2)_5\text{NH}_3)_2\text{PbI}_4 \cdot x\text{I}_2$ than $(\text{CH}_3(\text{CH}_2)_5\text{NH}_3)_2\text{PbI}_4 \cdot x\text{I}_2$.⁷ However, the exciton binding energy for these compounds were reduced upon intercalation, with a value of 180 meV being reported for $(\text{ICH}_2(\text{CH}_2)_5\text{NH}_3)_2\text{PbI}_4 \cdot x\text{I}_2$.⁷ The intercalation of DMSO and DMF into $(\text{PEA-OH})\text{PbBr}_4$ (where $\text{PEA-OH} = \text{HOC}_6\text{H}_5(\text{CH}_2)_2\text{NH}_3^+$) has also been studied.⁸ Here, the intercalation of DMSO was very stable, due to hydrogen bonds between the PEA-OH and DMSO.⁸ However, the changes in electronic structure were small and $(\text{PEA-OH})\text{PbBr}_4 \cdot \text{DMSO}$ also had a short carrier lifetime.⁸ It was also possible to intercalate DMF into $(\text{PEA-OH})\text{PbBr}_4$, and both $(\text{PEA-OH})\text{PbBr}_4 \cdot \text{DMF}$ and $(\text{PEA-OH})\text{PbBr}_4 \cdot 2\text{DMF}$ were reported.⁸ Variable quantities of DMF could be intercalated, which led to mixed phase materials being observed.⁸ In addition, we note that the single crystal structures published had high R-factors and large residual electron densities, which may indicate issues with the crystal structure or refinement.⁸ $(\text{PEA-OH})\text{PbBr}_4 \cdot \text{DMSO}$ also could be used as a photodetector.⁸

2. As per the theoretical calculations carried out here, the $[\text{H}_3\text{N}(\text{CH}_2)_m\text{NH}_3]\text{PbX}_4 \cdot \text{X}_2$ ($m = 7$) compounds are among the most stable ones with high negative E_{binding} energies. The most stable compound among all the intercalated compounds studied in this submission is $[\text{H}_3\text{N}(\text{CH}_2)_7\text{NH}_3]\text{PbI}_4 \cdot \text{Br}_2$ with the E_{binding} of -0.62 eV. However, the authors could not isolate this compound as single crystal or as thin film (Table S8), but still they have projected this as the most promising photovoltaic materials among all the compounds on page number 5. In this case the theoretical results and experimental observations contradict each other. Not only this, none of the other proposed compounds with $m = 7$ were accessible as thin films but were isolated as single crystals. Why are they unable to make their most stable compound?

We thank the reviewer for their comments on this point and also draw their attention to point four in Reviewer #2's comments, which stated that "*The authors may have missed the importance of even/odd alteration of the α - ω diamines. Odd chains pack poorly giving rise to less stable hybrid halides. They can potentially be more readily stabilized by X_2 intercalation, but also, the poorer chain packing may help admit X_2 .*"

As pointed out by Reviewer #2, it is well-known in the literature that there is a difference in the structure of organic inorganic metal halide layered perovskites when diammonium cations are used, which depends on whether the diammonium cations contain even or odd numbers of carbon atoms in their carbon backbone. For example, the work of Lemmerer clearly shows this (A. Lemmerer, D. G. Billing, *CrystEngComm*, 2012, 14, 1954). The work of Lightfoot *et al.* also shows the importance of odd/even number of carbon atoms in the carbon chain. (T. Li, A. J. Bradford S L. Lee P. Lightfoot, *Chem. Mater.* 2024, 36, 10, 5228–5237). Depending on whether the carbon chain contains an odd or even number of carbon atoms, the space groups adopted are either polar or centrosymmetric. In turn, this is reported to influence hydrogen bonding and the tilting of the inorganic layers with respect to one another.

The $m = 7$ parent materials, $[H_3N(CH_2)_7NH_3]PbX_4$ are known to be unstable. In agreement with studies in the literature, we also were not able to make single crystals of $[H_3N(CH_2)_7NH_3]PbX_4$. We believe that the high $E_{binding}$ of the intercalated $[H_3N(CH_2)_7NH_3]PbX_4 \cdot X_2$ could be driven by the instability of the parent $[H_3N(CH_2)_7NH_3]PbX_4$. This suggests that there is actually good agreement between computational and experimental. In addition, several experimental observations also provide further evidence of the instability of $[H_3N(CH_2)_7NH_3]PbX_4$. During our attempts to prepare $[H_3N(CH_2)_7NH_3]PbI_4 \cdot I_2$ using the route first based on the method by Poglitsch and Weber, which uses hydriodic acid, a competing phase, $[H_2N(CH_2)_7NH_3]_4Pb_3I_{12} \cdot 2I$ was identified, meaning that we could not prepare $[H_3N(CH_2)_7NH_3]PbI_4 \cdot I_2$ in a pure form. Instead, we had to use a completely different synthetic method to access pure $[H_3N(CH_2)_7NH_3]PbI_4 \cdot I_2$. This worked by preparing $[H_3N(CH_2)_7NH_3]I_2 \cdot I_2$, then carrying out a solid state reaction, grinding $[H_3N(CH_2)_7NH_3]I_2 \cdot I_2$ and PbI_2 . The mixture was heated to 160 °C for 1 hour and resulted in the formation of pure $[H_3N(CH_2)_7NH_3]PbI_4 \cdot I_2$. The fact that different synthetic conditions are required to prepare this material gives an insight into the thermodynamics and kinetics of the synthesis process. As phase pure $[H_3N(CH_2)_7NH_3]PbI_4 \cdot I_2$ was made at high temperatures of 160 °C, this suggests that a large activation energy must be overcome in order to form this product. Therefore, in the solution reactions, which occur at lower temperatures, the $[H_2N(CH_2)_7NH_3]_4Pb_3I_{12} \cdot 2I$ product is expected to be the kinetic product, with a lower activation energy barrier to overcome. The competition between the phases could be studied in more detail in

the future. We point out that the most successful way of preparing thin films of [H₃N(CH₂)₆NH₃]PbX₄·X₂ and [H₃N(CH₂)₈NH₃]PbX₄·X₂ was to firstly prepare a thin film of the parent precursor, [H₃N(CH₂)₆NH₃]PbX₄ and secondly, to expose that precursor to the vapour of the halogen which is desired for intercalation.

In addition, we note that in the publication by the Kanatzidis Group (X. Li, J. Hoffman, W. Ke, M. Chen, H. Tsai, W. Nie, A. D. Mohite, M. Kepenekian, C. Katan, J. Even, M. R. Wasielewski, C. C. Stoumpos and M. G. Kanatzidis, *J. Am. Chem. Soc.* 2018, 140, 12226–12238) which discusses the preparation of materials with composition (NH₃C_mH_{2m}NH₃)(CH₃NH₃)_{n-1}Pb_nI_{3n+1} ($m = 4-9$; $n = 1-4$), no single crystal structures of the $m = 7, n = 1$ compound are reported. Although PXRD data were obtained for the $m = 7$ materials, Rietveld refinement was not reported in the manuscript, which could have been used to prove that the $m = 7, n = 1$ material had indeed been structurally characterised. Therefore, it is clear that the $m = 7$ compound is extremely difficult to make as our group and the Kanatzidis group, which is a leading group working in this field, have not currently reported the single crystal structure of this compound, which would enable a full characterisation of the crystal structure. This supports the theory that the $m = 7$ compounds are unstable.

We note that in the main text of our manuscript we state that "Experimental studies have shown that the $m = 7$ [H₃N(CH₂)₇NH₃]PbX₄ parent structures are unstable and none could be prepared in crystal form."^{10,11}

Also, In the paper "C. Deng, G. Zhou, D. Chen, J. Zhao, Y. Wang, and Q. Liu, *J. Phys. Chem. Lett.* 2020, 11, 2934–2940", the compound (C₇H₁₈N₂)PbBr₄ [where (C₇H₁₈N₂) is 1,7-diammoniumheptane]" was reported. However, in this paper, again only a polycrystalline sample of (C₇H₁₈N₂)PbBr₄ was prepared, which precluded analysis via single crystal X-ray diffraction. Structural models were built, based on the $m = 9$ structures reported by the Kanatzidis group, rather than a full structure solution for the $m = 7$ compound from powder diffraction data.

3. "A total of 54 compounds were screened, with eleven compositions predicted to exist, having E_{binding} less than -0.16 eV." What is so special about -0.16 eV?

We thank the reviewers for drawing our attention to this. As far as we understand, there is nothing special about the -0.16 eV, so we have deleted "having E_{binding} less than -0.16 eV".

4. Authors have written "excellent agreement between the experimental PXRD pattern collected at room temperature" however I find that the matching of experimental and simulated patterns is poor in some of the latter members of $[\text{H}_3\text{N}(\text{CH}_2)_m\text{NH}_3]\text{PbX}_4\cdot\text{X}_2$ family. Figure S10- $\text{H}_3\text{N}(\text{CH}_2)_8\text{NH}_3]\text{PbBr}_4\cdot\text{I}_2$: Position mismatch for the peaks at $2\theta = \sim 15^\circ, \sim 22^\circ \sim 29^\circ, \sim 33^\circ$. Here, I have mentioned only the tall peaks, but the mismatch is there in almost all the higher 2θ peaks. Similarly Figure S11, $[\text{H}_3\text{N}(\text{CH}_2)_8\text{NH}_3]\text{PbI}_4\cdot\text{I}_2$, the peaks at $2\theta = \sim 15^\circ, \sim 22^\circ \sim 29^\circ$ are mismatching with the simulated pattern. How can authors say that these are excellent matches? Author should redo these measurement or explain the differences.

Fitting the PXRD data using Le Bail, Pawley or Rietveld refinement is a much better way of showing that the single crystal X-ray diffraction data match up with PXRD data. We note that single crystal X-ray diffraction data were collected at 173 K and the PXRD data were collected at 293 K, so we expect some increases in unit cell parameters (and hence shift in peak position) between the simulated and experimental PXRD data. In addition, we note that only phase identification quality PXRD data were collected, rather than Rietveld quality data. As a result, in most cases, we were unable to reliably refine realistic atomic displacement parameters when performing Rietveld refinement. For some samples, we also encountered some problems in fitting the data due to preferred orientation or poor powder averaging statistics. The samples are prepared in highly crystalline forms, but due to the organic-inorganic metal halides being much softer than other inorganic materials, such as inorganic oxides which are typically characterised by PXRD, grinding of samples before PXRD data collection can result in a degradation of crystallinity or deintercalation. As a result, sometimes we have PXRD data which show preferred orientation or poor powder averaging statistics, which can make it difficult to fit the data. Another factor which can cause difficulties in fitting the PXRD data (collected at room temperature) is the possibility of phase transitions in the materials between 173 K (temperature used for single crystal data collection) and 298 K (temperature used for PXRD data collection). We also note that in some samples an iron peak was observed in the PXRD pattern due to the sample holder. Fits to the PXRD data have now been given in the supplementary information and are also given below.

Figure R1: Rietveld fit for $[\text{H}_3\text{N}(\text{CH}_2)_7\text{NH}_3]\text{PbBr}_4\cdot\text{Br}_2$. $a = 8.0259(3) \text{ \AA}$, $b = 8.2729(4) \text{ \AA}$, $c = 29.0141(13) \text{ \AA}$ and $\text{volume} = 1926.49(15) \text{ \AA}^3$. $R_{\text{wp}} = 11.795\%$. 12 background parameters, a term for peak asymmetry, profile parameters, sample displacement, unit cell parameters, atomic coordinates and individual U_{iso} for Pb and Br were refined.

Figure R2: Pawley fit for $[\text{H}_3\text{N}(\text{CH}_2)_7\text{NH}_3]\text{PbBr}_4\cdot\text{lBr}$. $a = 8.03974(18) \text{ \AA}$, $b = 29.4684(6) \text{ \AA}$, $c = 8.1699(2) \text{ \AA}$ and $\text{volume} = 1935.61(8) \text{ \AA}^3$. $R_{\text{wp}} = 7.779\%$. Pawley refinement, where 12 background parameters, asymmetry, profile parameters, sample displacement and unit cell parameters were refined.

Figure R3: Pawley fit for $[H_3N(CH_2)_7NH_3]PbI_4 \cdot I_2$. $a = 8.5223(3) \text{ \AA}$, $b = 8.5642(3) \text{ \AA}$, $c = 31.1607(5) \text{ \AA}$ and volume = $2274.34(12) \text{ \AA}^3$. $R_{wp} = 10.604\%$. 12 background parameters, asymmetry, profile parameters, sample displacement, spherical harmonics for preferred orientation and unit cell parameters were refined.

Figure R4: Rietveld fit for $[H_3N(CH_2)_8NH_3]PbBr_4 \cdot I_2$. $a = 8.2762(9) \text{ \AA}$, $b = 8.3066(11) \text{ \AA}$, $c = 30.379(3) \text{ \AA}$, $\beta = 92.510(15)^\circ$ volume = $2086.4(4) \text{ \AA}^3$. $R_{wp} = 12.485\%$. 12 background parameters, asymmetry, profile parameters, sample displacement, unit cell parameters and atomic coordinates for Pb, Br and I were refined.

Figure R5: Rietveld fit for $[H_3N(CH_2)_8NH_3]PbI_4 \cdot I_2$. $a = 8.8161(3) \text{ \AA}$, $b = 8.7167(3) \text{ \AA}$, $c = 30.1314(11) \text{ \AA}$, $\beta = 92.530(4)^\circ$, volume = $2313.28(16) \text{ \AA}^3$, $R_{wp} = 6.647\%$. 12 background parameters, a term for peak asymmetry, profile parameters, sample displacement, unit cell parameters, atomic coordinates and individual U_{isots} for Pb and I were refined.

Figure R6: Rietveld fit for $[H_3N(CH_2)_9NH_3]PbI_4 \cdot I_2$. $a = 30.2366(10) \text{ \AA}$, $b = 9.0411(4) \text{ \AA}$, $c = 8.8124(4) \text{ \AA}$, $\beta = 89.576(4)^\circ$, volume = $2409.01(16) \text{ \AA}^3$, $R_{wp} = 6.687\%$. 12 background parameters, a term for peak asymmetry, profile parameters, sample displacement, unit cell parameters, atomic coordinates and individual U_{isots} for Pb and I were refined.

5. Optical absorption spectra are confusing. Authors have written...“Of the bulk samples synthesised, $[\text{H}_3\text{N}(\text{CH}_2)_7\text{NH}_3]\text{PbI}_4\cdot\text{I}_2$ had the lowest band gap (1.74 eV)”, which does not seem correct to me. From the UV spectra (Figure S17), it is clear that the bandgaps of $[\text{H}_3\text{N}(\text{CH}_2)_m\text{NH}_3]\text{PbI}_4\cdot\text{I}_2$, are in the order $m = 9 < m = 7 < m = 8$. But authors have listed in Table S7 in the sequence of $m = 7 < m = 8 < m = 9$. How were the bandgaps determined? They should reassess the bandgap values.

We thank the reviewer for pointing out the issue in determining the bandgap value of the compounds, especially $[\text{H}_3\text{N}(\text{CH}_2)_9\text{NH}_3]\text{PbI}_4\cdot\text{I}_2$. We have now updated Figure S17, including the baseline and absorption edge line. In the submitted manuscript, we reported the bandgap of $[\text{H}_3\text{N}(\text{CH}_2)_9\text{NH}_3]\text{PbI}_4\cdot\text{I}_2$ as 1.94 eV, as the K-M plot of that sample shows two parts having similar gradient. We now corrected the bandgap value of $[\text{H}_3\text{N}(\text{CH}_2)_9\text{NH}_3]\text{PbI}_4\cdot\text{I}_2$ to 1.83 eV, so the current sequence change to: $m = 7$ (1.77 eV) < $m = 9$ (1.83 eV) < $m = 8$ (1.94 eV).

6. Discussion on the photophysical properties needs to be strengthened as most of the information provided is about their “observations” rather than comparison to the established literature.

We thank the reviewer for suggestion to improve the photophysical studies. The section on photoluminescence studies has been rewritten. The conclusions from each result and corresponding Figures have been discussed. Additional references have also been added to the manuscript. We have now also performed a comprehensive photoluminescence excitation (PLE) experiment on the $[\text{H}_3\text{N}(\text{CH}_2)_8\text{NH}_3]\text{PbI}_4$ (Figure 5c and 5d) and $[\text{H}_3\text{N}(\text{CH}_2)_8\text{NH}_3]\text{PbI}_4\cdot\text{I}_2$ (Figure 5e and 5f) perovskite at 4 K temperature, after changes to the experimental setup that allowed simultaneous broadband detection spanning the ultraviolet to near-infrared spectrum (please see also Response to Reviewer 2, point 6).

7. Page 13, lines 17 and 18: the sequence ($m = 8 < m = 7 < m = 9$) of red emission spectra is provided in “wavelength” unit, while that of the absorption spectra in “Energy” unit ($m = 7 < m = 8 < m = 9$). For easy comparison, the sequences need to be in the same measurement unit (i.e. “wavelength” or “energy”). Similarly, for the sequence provided on line 8 of this page about the green emission $m = 8 < m = 7 < m = 9$).

The text has now been amended to allow better clarity of the trend. It now reads as:

‘Interestingly, the broadband red emission of the intercalated perovskites (Figure 5b) exhibits a different sequence and longer emission wavelength, corresponding to smaller bandgap, with increasing m ($m = 7 < m = 8 < m = 9$). This sequence contradicts the sequence for bandgaps determined at room temperature from UV-Visible spectroscopy, ($m = 7 < m = 9 < m = 8$), which suggests that attributing the broadband emission solely to the new CBM emission oversimplifies this phenomenon.’

8. Why are the measurements carried out at particularly 4 K?

We apologise if the temperature of different measurements were unclear. All our cryogenic measurements were performed using a liquid helium continuous flow cryostat. Since the boiling point of liquid helium is 4 K, it is easier to cool all the 7 intercalated samples directly to 4K. One parent and its intercalated crystal sample was then chosen to perform the variable temperature dependent study.

9. How did authors confirm that in $[\text{H}_3\text{N}(\text{CH}_2)_7\text{NH}_3]\text{PbBr}_4\cdot\text{IBr}$, the intercalant has I and Br in 1:1 ratio rather than an alloyed variant of IBr with more iodine content than bromine. The I-Br bond distance in the CIF is close to that of the I-I distance of I_2 molecules and far from the Br-Br distance of Br_2 molecule.

We thank the reviewer for this question. Firstly, we point out that the compound is that was made using IBr, so $[\text{H}_3\text{N}(\text{CH}_2)_7\text{NH}_3]\text{PbBr}_4\cdot\text{IBr}$ it is what would be expected in the first instance. Of course, IBr does have some tendency towards homolytic dissociation which would give a mix of I_2/Br_2 . However, one piece of evidence that we have IBr in our compound, rather than a I_2 or Br_2 comes from our thermal gravimetric analysis. Upon heating, we would expect a mass loss corresponding to one molecule of IBr per formula unit of the parent perovskite, $[\text{H}_3\text{N}(\text{CH}_2)_7\text{NH}_3]\text{PbBr}_4$. This would correspond to a mass loss of 23.9%. Mass loss of one molecule of Br_2 per formula unit would correspond to a mass loss of 19.5% and mass loss of one molecule of I_2 per formula unit would correspond to a mass loss of 27.8%. From our thermogravimetric analysis, which was shown in Figure S16, we can see that the mass loss most closely corresponds to the 24.4%, which is closest to the value expected when IBr is lost from the crystal structure.

A search of the CSD for I-Br distances for compounds that contain discrete IBr show up relatively close to the distance for I_2 . However, we note that there are only five structures with discrete IBr, with bond distances ranging between 2.58 and 2.72 Å. For the $[\text{H}_3\text{N}(\text{CH}_2)_7\text{NH}_3]\text{PbBr}_4\cdot\text{IBr}$ structure reported in our work, the I-Br distance is 2.67 Å which fits nicely with the CSD data for structures containing discrete IBr. As the reviewer correctly mentions, this does indeed also match up with the common I-I range for discrete I_2 in structures (which have bond distances ranging from 2.62 to 2.89 Å). From our refinements using single crystal X-ray diffraction behaviour, no unexpectedly large or small anisotropic displacement parameters are observed for IBr, which if present could have suggested the presence of I_2 or Br_2 in the structure. Therefore based on our TGA results and existing structures published in the CSD, we are confident that IBr is indeed present in the structure of $[\text{H}_3\text{N}(\text{CH}_2)_7\text{NH}_3]\text{PbBr}_4\cdot\text{IBr}$.

Reviewer #2 (Remarks to the Author):

The manuscript by Payne and coworkers discusses the intercalation of neutral, molecular X₂ species into 2D hybrid halide perovskites as a means of tuning the electronic structure. The manuscript requires major revisions:

We thank the reviewer for this positive response. We will address your suggestions for improvements in the text below.

1. The title is long and the first clause is somewhat pointless.

We thank the reviewer for this suggestion. We have changed the title to "Advancing Intercalating Strategies in Layered Hybrid Perovskites by using Synthesis and Simulations".

2. The authors do a poor job of introducing the subject of intercalation of molecules into 2D perovskites, including molecular dihalogens. They done a better job of this in their previous publication on the topic (reference 7) but the prior work of others is ignored here completely.

We apologise for lack of background on intercalation in layered halide perovskites and note that Reviewer #1 made a very similar comment. We are pleased that the Reviewer found the background in our previous paper useful. Of course, we should cite appropriate papers and we have modified the introduction accordingly. As noted above, we have also added the reference suggested by Reviewer #1 (*Adv. Mater. Technol.* 2023, 8, (5), 2201465).

The new text now reads:

In 1986, Maruyama *et al.* reported that small molecules, including 1-chloronaphthalene, *o*-dichlorobenzene and hexane, could be reversibly intercalated into layered hybrid perovskites (C₁₀H₂₁NH₃)₂CdCl₄ and (C₉H₁₉NH₃)₂PbI₄.¹ To the best of our knowledge, this was the first report of intercalation in layered hybrid perovskites. However, in this study, single crystals of the intercalated compound were not obtained and only changes in unit cell parameters could be observed.¹ Mitzi *et al.* then looked at the intercalation of C₆H₆ and C₆F₆ into (C₆F₅C₂H₄NH₃)₂SnI₄ and (C₆H₅C₂H₄NH₃)₂SnI₄ respectively.² In this instance, intercalation of C₆F₆ into (C₆H₅C₂H₄NH₃)₂SnI₄ only resulted in an 0.04 eV change in band gap, despite the distance between the [SnI₄]_∞ layers changing from 16.3 Å to 20.6 Å.² More recently, intercalation has played an important role in the processing of organic-inorganic metal halides, as solvents such as DMF etc have been postulated to intercalate between PbI₂ layers.³⁻⁵ Nag has also looked at intercalation in a number of compounds, including (BA)₂PbI₄ (where BA = butylammonium), and (PEA)₂PbI₄ (where PEA = phenylethylammonium), but we note that no single crystal structures were obtained from single crystal X-ray diffraction.⁶ In this work, (BA)₂PbI₄ displayed two peaks in the photoluminescence spectrum, which was attributed to two different areas of the crystal (edge and terrace), which suggested electronic interactions between neighbouring [PbI₄]_∞ layers.⁶ When iodine was intercalated, only a single emission was observed in the photoluminescence and this was found at higher energies.⁶ In this study, the lower energy peak had been attributed to edge emission. This process was reversible. The same group then went to look at hexane intercalation

into (DA)₂PbI₄ (where DA = decyl ammonium), which again changed the PL emission from dual to single emission.⁶ However, the intercalated molecules were prone to deintercalation, which precluded the growth of crystals suitable for single crystal X-ray diffraction studies.⁶ As a result, (PEA)₂SnI₄·C₆F₆, previously prepared by Mitzzi *et al.* was investigated.^{2,6} It also showed dual emission in the PL spectra and like the other compounds, the low energy PL emission disappeared upon intercalation of the C₆F₆ molecule.⁶ To complete the study, Nag *et al.* also looked at intercalation in (C_mH_{2m+1}NH₃)₂PbI₄, where the length of the carbon chain was systematically varied.⁶ As the carbon chain length increased, the PL went from dual emission to single emission, with the loss of the low energy peak.⁶ Karunadasa looked at the intercalation of I₂ into (CH₃(CH₂)₅NH₃)₂PbI₄ and the related compound containing a terminal alkyl iodide group, (ICH₂(CH₂)₅NH₃)₂PbI₄.⁷ In these compounds, I₂ was only stable for a short time and no single crystal XRD could be obtained for either material, preventing full structural characterisation of these materials.⁷ We note that the intercalation of I₂ was found to be more stable in (ICH₂(CH₂)₅NH₃)₂PbI₄·xI₂ than (CH₃(CH₂)₅NH₃)₂PbI₄·xI₂.⁷ However, the exciton binding energy for these compounds were reduced upon intercalation, with a value of 180 meV being reported for (ICH₂(CH₂)₅NH₃)₂PbI₄·xI₂.⁷ The intercalation of DMSO and DMF into (PEA-OH)PbBr₄ (where PEA-OH = HOC₆H₅(CH₂)₂NH₃⁺) has also been studied.⁸ Here, the intercalation of DMSO was very stable, due to hydrogen bonds between the PEA-OH and DMSO.⁸ However, the changes in electronic structure were small and (PEA-OH)PbBr₄·DMSO also had a short carrier lifetime.⁸ It was also possible to intercalate DMF into (PEA-OH)PbBr₄, and both (PEA-OH)PbBr₄·DMF and (PEA-OH)PbBr₄·2DMF were reported.⁸ Variable quantities of DMF could be intercalated, which led to mixed phase materials being observed.⁸ In addition, we note that the single crystal structures published had high R-factors and large residual electron densities, which may indicate issues with the crystal structure or refinement.⁸ (PEA-OH)PbBr₄·DMSO also could be used as a photodetector.⁸

3. The layered perovskites studied here are clearly of the Dion-Jacobson type, but the introduction only discusses R-P phases.

We thank the Reviewer for raising this point. However, based on our crystallographic work, this is not strictly true. Whether a layered perovskite forms a Dion-Jacobson structure or Ruddlesden Popper structure depends on the displacement of adjacent inorganic [PbX₄]_∞ layers with respect to one another. This is defined by the layer shift factor (**L_s**), as discussed in the main article. For ideal Ruddlesden-Popper phases, the **L_s** is (0.5,0.5), whereas for Dion Jacobson phases, the **L_s** is (0.0,0.0). As there are such a variety of organic ammonium cations which may be incorporated into layered organic-inorganic perovskites, the layer shift factor can have intermediate values, which are intermediate to those expected for a Ruddlesden-Popper structure or a Dion-Jacobson structure. Hence for example, [H₃N(CH₂)₆NH₃]PbBr₄·Br₂ has a **L_s** of (0.13,0.13) and therefore a 'Dion-Jacobson like' structure (as it is closer to (0.0, 0.0)), whereas [H₃N(CH₂)₆NH₃]PbBr₄ has a **L_s** of (0.39, 0.39) which means that the it has 'Ruddlesden-Popper like' structure (as it is closer to (0.5, 0.5)). In order to clarify this to the reader and combined with the comment 4 from the Reviewer. we have now updated our new Figure 2 in the main article and highlighted the **L_s** for each structure. We also direct the reviewers to the paper by J. A. McNulty and P. Lightfoot, *IUCr*, 2021, 8, 485–513.

4. Nowhere in the manuscript is a complete structure displayed that shows the 2D perovskite slabs with the amines and the X_2 intercalant. This would be helpful to the reader, and one of the figures from the SI could be moved.

We thank the Reviewer for raising this point and agree that it would be very useful. We note that some full structures were shown in the supplementary information. Therefore, we have moved Figures S13–S15 to the main article as Figures 2b-d, showing the full structure of all the intercalated perovskites.

5. The authors may have missed the importance of even/odd alteration of the α - ω diamines. Odd chains pack poorly giving rise to less stable hybrid halides. They can potentially be more readily stabilized by X_2 intercalation, but also, the poorer chain packing may help admit X_2 .

The trend in the PL also reflects this.

We thank Reviewer #2 for this very important point. This is certainly true and has been discussed in the literature (A. Lemmerer, D. G. Billing, *CrystEngComm*, 2012, 14, 1954, and X. Li, J. Hoffman, W. Ke, M. Chen, H. Tsai, W. Nie, A. D. Mohite, M. Kepenekian, C. Katan, J. Even, M. R. Wasielewski, C. C. Stoumpos and M. G. Kanatzidis, *J. Am. Chem. Soc.* 2018, 140, 12226–12238). We should have cited some appropriate papers on this matter so apologise for this oversight. We have now added in the following reference. We note that a Reviewer #1 also raised a similar issue- so please see our response to Reviewer #1 (point 2)

We also have a calculation on parent perovskites showing that the parent perovskite with odd numbers of carbon atoms has a higher energy (Figure R7). This figure can help to demonstrate the odd/even effect in the number carbon atoms in the backbone of the $[H_3N(CH_2)_mNH_3]^{2+}$ cations.

Figure R7: The black line represents the total energy obtained from DFT calculations. Since the number of atoms in each parent perovskite differs, the energy has been linearly fitted based on the carbon chain length. The red values indicate the energy deviations above the linear fit, with higher values suggesting

greater instability. The observed trend shows that the odd-numbered carbon chain parents generally have energies higher than the linear fit, indicating relative instability.

We also note that, based on our experimental results, the appropriate range for intercalation is for $[\text{H}_3\text{N}(\text{CH}_2)_m\text{NH}_3]^{2+}$ cations, where $m = 6 - 9$. Only the largest iodine molecules show one example of intercalation at $m = 9$. Additionally, the $m = 7$ is compounds are significantly higher in energy than other layered perovskites made with $[\text{H}_3\text{N}(\text{CH}_2)_m\text{NH}_3]^{2+}$ cations which have odd numbers of carbon atoms in their backbones, including $m = 3, 5,$ and 9 .

6. What is the excitation wavelength for the PL. The PL has a broad band emission that is attributed to the intercalant. Can they excite into the board band directly? PL excitation spectra may be informative. The explanation of the breadth is incomprehensible. Please note that a molecular species that is not held strongly, is very likely to have the phonon landscape that would lead top broad emission. On this point, STEs are referred to as being extrinsic to the material. They are not.

We fully agree that the STEs are not extrinsic to the material. We have corrected this in the main article. The excitation wavelength for the PL studies were done using a 415 nm constant wavelength laser. This detail was mentioned in the supplementary text earlier. It has now been added to the main article.

In our earlier experiments, no broadband PL was detected on exciting directly into the intercalated band (500 – 550 nm). We have now performed a comprehensive photoluminescence excitation (PLE) experiment on $[\text{H}_3\text{N}(\text{CH}_2)_8\text{NH}_3]\text{PbI}_4$ (Figure R8a and R8b) and $[\text{H}_3\text{N}(\text{CH}_2)_8\text{NH}_3]\text{PbI}_4 \cdot \text{I}_2$ (Figure R8c and R8d) at a temperature of 4 K, after changes to the experimental setup that allowed simultaneous broadband detection, spanning the ultraviolet to near-infrared spectrum. The experimental setup was upgraded to allow tuneable pulsed excitation using an NKT Supercontinuum laser. The broadband emission from the perovskite sample implied that the high resolution narrow range spectrometer used previously was not required. So the setup was complemented with a fibre coupled large bandwidth spectrometer. This allowed simultaneous detection of the narrow band green emission and the broadband red emission. PLE spectra were recorded by tuning the excitation wavelength in steps of 2 nm all the way from 400 nm to 540 nm to detect broadband emission at wavelengths larger than 550 nm, and from 400 nm to 440 nm detect narrow band emission at wavelength larger than 450 nm, using 550 nm and 450 nm edge pass filters in the excitation and detection arms of the optical setup. Example spectra are plotted in Figure R9, where emission from the FE and the intercalated STE band was detected (with a 450 nm long pass filter) upon excitation at 440 nm (with a 450 nm short pass filter), and emission from the intercalated STE band detected (with a 550 nm long pass filter) upon excitation at 480 nm (with a 550 nm short pass filter).

In this result, there are four exciton bands in both $[\text{H}_3\text{N}(\text{CH}_2)_8\text{NH}_3]\text{PbI}_4$ and $[\text{H}_3\text{N}(\text{CH}_2)_8\text{NH}_3]\text{PbI}_4 \cdot \text{I}_2$, with peak absorption of the first three bands at 410 nm, 440 nm and 480 nm. The FE has an absorption peak at 480 nm and emission peak at 490 nm.

Emission from the two higher exciton states was not observed perhaps due to ultrafast relaxation into the FE exciton. While red emission from the broadband was observed with all excitation wavelengths from 400 nm to 540 nm, the maximum intensities were obtained when excited directly into the three higher exciton bands. From the PLE spectra of the parent perovskite (Figure R8(a) and R8(b)), it is evident that the broadband is populated by relaxation from the FE. This shows that the broadband emission in the parent perovskite does not arise from a permanent defect state and instead forms an intrinsic STE state. On the other hand, while the intercalated perovskite largely retains its original band structure of the three higher excitonic states, the PLE intensities are altered relative to one another compared to the parent perovskite (Figure R8(a) and R8(b)). While a clear absorption peak of the broadband emission at energies lower than the FE is not observed, there is a non-negligible absorption below the FE. From this result we can infer that intercalation modifies the nature of the STE due to modification of the band structure. This result is also supported by band structure calculations.

We thank the reviewer for the suggestions and have included this results in the main text.

Figure R8 (a) Photoluminescence excitation spectra of $[H_3N(CH_2)_8NH_3]PbI_4$ measured at 4K. (b) Combined emission spectra, showing contributions from both narrow-band and broad-band emissions of $[H_3N(CH_2)_8NH_3]PbI_4$, recorded with excitation wavelengths ranging from 400 to 540 nm. (c)

Photoluminescence excitation spectra of $[\text{H}_3\text{N}(\text{CH}_2)_8\text{NH}_3]\text{PbI}_4 \cdot \text{I}_2$ measured at 4K. (d) Combined emission spectra, showing contributions from both narrow-band and broad-band emissions of $[\text{H}_3\text{N}(\text{CH}_2)_8\text{NH}_3]\text{PbI}_4 \cdot \text{I}_2$, recorded with excitation wavelengths ranging from 400 to 540 nm.

Figure R9: Photoluminescence excitation spectra collected from the $[\text{H}_3\text{N}(\text{CH}_2)_8\text{NH}_3]\text{PbI}_4$ using 440 nm and 480 nm excitation.

7. In the conclusion, the board-band emission is due to STEs but this is in seeming contradiction with the previous discussion.

We apologise for this contradiction. As discussed above, intercalation allows modification of the conduction band minima such that while the STE is populated largely by relaxation from the FE, a weak direct excitation of the intercalated-STE band is possible.

We have now reworded the sentence as follows:

"Photoluminescence studies of the intercalated perovskite samples show that intercalation allows tuneability of the STE energy. Intercalation also modifies the nature of the STE to permit direct, albeit a weak, excitation into the intercalated band."

8. On the whole, the quality of figures is poor. There is extensive use of bar graphs that make no sense and these are difficult to read. These are all cases where scatter plots make more sense. Also, figures with lots of too-small text are never appealing.

We thank Reviewer #2 for this comment. We have now modified the Figures in the manuscript. The bar charts have been changed to scatter plots.

Reviewer #3 (Remarks to the Author):

We value Nature Communication's initiative to train Early Career researchers in peer review and thank Reviewer #3 for their time, effort and careful consideration whilst reviewing the manuscript.

Summary of changes to the manuscript

All changes to the manuscript and supplementary information have been highlighted and uploaded for review. A summary of the changes is also given below.

1. The crystallography studies section has been optimised. The original **Figure S13 to S15** have been included into **Figure 2(b) to 2(d)** in the Main Text, for showing the full crystal structures. The original **Figures 2(c)** and **2(b)** have been converted into scatter plots and moved to the Supplementary Information as **Figures S13** and **S14**, respectively. The discussion on the crystal structures, particularly the L_s values of the intercalated perovskites, has been integrated into the main text.

Figure 2. (a) Part of the crystal structure of seven intercalated perovskites ($[\text{H}_3\text{N}(\text{CH}_2)_m\text{NH}_3]\text{PbX}_2 \cdot \text{X}_2$). I, Br, N, and C atoms are represented by purple, brown, blue, and black spheres respectively, whilst the Pb-centred polyhedra are shown in pink. Hydrogen atoms are omitted for clarity. The linear 'length' of the $[\text{H}_3\text{N}(\text{CH}_2)_m\text{NH}_3]^{2+}$ cation is labelled. (b) Two views of the crystal structures for all three $m = 7$ samples, including $[\text{H}_3\text{N}(\text{CH}_2)_7\text{NH}_3]\text{PbBr}_4 \cdot \text{Br}_2$, $[\text{H}_3\text{N}(\text{CH}_2)_7\text{NH}_3]\text{PbBr}_4 \cdot \text{IBr}$ and $[\text{H}_3\text{N}(\text{CH}_2)_7\text{NH}_3]\text{PbI}_4 \cdot \text{I}_2$. (c) Two views of the crystal structure for all three $[\text{H}_3\text{N}(\text{CH}_2)_m\text{NH}_3]\text{PbI}_4 \cdot \text{I}_2$ ($m = 7, 8$ and 9) intercalated samples. (d) : Two views of the crystal structure for the two $m = 8$ samples $[\text{H}_3\text{N}(\text{CH}_2)_8\text{NH}_3]\text{PbBr}_4 \cdot \text{I}_2$ and $[\text{H}_3\text{N}(\text{CH}_2)_8\text{NH}_3]\text{PbI}_4 \cdot \text{I}_2$.

Figure S13: Comparisons between computational and experimental values obtained for the key structural parameters: θ_1 , θ_2 (scatter with reference to left x-axis) and $|\Delta D|$ (scatter with reference to right x-axis).

Figure S14: Structural parameters D_h (scatter with reference to left y-axis) and D_L (scatter with reference to right y-axis,) for seven intercalated perovskites from analysis of single crystal XRD data.

- A new **Figure S15**, showing the single octahedra of [H₃N(CH₂)₈NH₃]PbBr₄·I₂, is now given. This figure provides a clearer visual reference for readers, helping them better understand the impact of size-mismatch intercalation on octahedral distortion.

Figure S15: Individual [PbBr₄]²⁻ octahedra of [H₃N(CH₂)₈NH₃]PbBr₄·I₂. The axial Pb-Br bonds tilt in opposite directions to accommodate the intercalation of I₂ molecules.

3. **Figure S17** has been improved, and the bandgap values have been reassessed.
4. **Figure 3(a), 3(b), 3(d) and 3(e)** have been converted to scatter plots for clarity.
5. The photoluminescence studies section with improved discussion has been rewritten.
6. The figures of PL spectra in the supplementary document have been consolidated to group similar properties together. **Figures S27 to S34** have now been merged into **Figures S27 to S29**. Consequently, **Figures S35 to S40** have been renumbered to **Figures S31 to S35**.

Figure S27: Narrowband photoluminescence of (a) [H₃N(CH₂)₈NH₃]PbI₄ and [H₃N(CH₂)₈NH₃]PbI₄·I₂ crystals at 4K, (b) [H₃N(CH₂)₈NH₃]PbI₄ and [H₃N(CH₂)₈NH₃]PbI₄·I₂ crystals at room temperature, and (c) [H₃N(CH₂)₈NH₃]PbI₄ and [H₃N(CH₂)₈NH₃]PbI₄·I₂ thin films at room temperature.

Figure S28: Broadband photoluminescence of (a) $[\text{H}_3\text{N}(\text{CH}_2)_8\text{NH}_3]\text{PbI}_4$ and $[\text{H}_3\text{N}(\text{CH}_2)_8\text{NH}_3]\text{PbI}_4 \cdot \text{I}_2$ crystals at 4K; (b) $[\text{H}_3\text{N}(\text{CH}_2)_8\text{NH}_3]\text{PbBr}_4$ and $[\text{H}_3\text{N}(\text{CH}_2)_8\text{NH}_3]\text{PbBr}_4 \cdot \text{I}_2$ Crystal at 4K; $[\text{H}_3\text{N}(\text{CH}_2)_6\text{NH}_3]\text{PbBr}_4$ and $[\text{H}_3\text{N}(\text{CH}_2)_6\text{NH}_3]\text{PbBr}_4 \cdot \text{Br}_2$ crystals at 4K

Figure S29 : (a) Photoluminescence broad peak of $[\text{H}_3\text{N}(\text{CH}_2)_7\text{NH}_3]\text{PbBr}_4 \cdot \text{Br}_2$ and $[\text{H}_3\text{N}(\text{CH}_2)_6\text{NH}_3]\text{PbBr}_4 \cdot \text{Br}_2$ crystals at 4K, (b) Photoluminescence broad peak of $[\text{H}_3\text{N}(\text{CH}_2)_7\text{NH}_3]\text{PbBr}_4 \cdot \text{Br}_2$ and $[\text{H}_3\text{N}(\text{CH}_2)_7\text{NH}_3]\text{PbBr}_4 \cdot \text{I}_2$ crystals at 4K

7. A new figure detailing results of the photoluminescence excitation (PLE) spectra collected on tuning the laser excitation wavelength with the parent and intercalated samples at 4K has been added to the main text as **Figure 5(c) – 5(f)**. The new **Figure 5** is shown below. Additional description about the PLE experimental setup was added to the supplementary document as given below.

For photoluminescence excitation spectra, a NKT Supercontinuum laser with tuneable pulsed excitation at 80MHz repetition rate, was used to excite the sample at 4K. A 450 nm (550 nm) short-pass filter was used to clean the laser and another 450 nm (550 nm) long pass filter was used in the detection arm to block the laser while collecting emission from the narrow band (broad band).

Figure 5: (a) Comparison of the sharp photoluminescence peak of $[H_3N(CH_2)_7NH_3]PbI_4 \cdot I_2$, $[H_3N(CH_2)_8NH_3]PbI_4 \cdot I_2$ and $[H_3N(CH_2)_9NH_3]PbI_4 \cdot I_2$ crystals at 4K; (b) Comparison of the broad photoluminescence peak of $[H_3N(CH_2)_7NH_3]PbI_4 \cdot I_2$, $[H_3N(CH_2)_8NH_3]PbI_4 \cdot I_2$ and $[H_3N(CH_2)_9NH_3]PbI_4 \cdot I_2$ crystals at 4K; (c) Photoluminescence excitation spectra of $[H_3N(CH_2)_8NH_3]PbI_4$ measured at 4K. (d) Combined emission spectra, showing contributions from both narrow-band and broad-band emissions of $[H_3N(CH_2)_8NH_3]PbI_4$, recorded with excitation wavelengths ranging from 400 to 540 nm. (e) Photoluminescence excitation spectra of $[H_3N(CH_2)_8NH_3]PbI_{4-x}I_2$ measured at 4K. (f) Combined emission

spectra, showing contributions from both narrow-band and broad-band emissions of $[\text{H}_3\text{N}(\text{CH}_2)_8\text{NH}_3]\text{PbI}_4 \cdot \text{I}_2$, recorded with excitation wavelengths ranging from 400 to 540 nm.

8. The original **Figure 5c** and **5d** are moved to SI, as the **Figure S30**.

Figure S30: (a) Variable temperature photoluminescence of $[\text{H}_3\text{N}(\text{CH}_2)_8\text{NH}_3]\text{PbI}_4$; (b) Variable temperature photoluminescence of $[\text{H}_3\text{N}(\text{CH}_2)_8\text{NH}_3]\text{PbI}_4 \cdot \text{I}_2$

9. Original text from the photoluminescence studies in the supplementary document has been changed from 'We note that considering the spatial arrangement of molecular orbitals in real space, excitation from the X-site anion p -orbitals to lead p -orbitals allow electrons to move into the halogen bond. Therefore, exciting the intercalated band is very weak. We used a green cw laser (530 nm) to activate $[\text{H}_3\text{N}(\text{CH}_2)_8\text{NH}_3]\text{PbI}_4 \cdot \text{I}_2$ at 4 K and no broadband emission was

observed. This is in contrast to what was observed using the 415 nm laser, *vide infra*. However, using a high energy tuneable pulsed laser, weak emission from the broadband was observed with excitation up to 540 nm. Figure S27-S30 show the results from PL experiments.' To 'We note that considering the spatial arrangement of molecular orbitals in real space, excitation from the X-site anion *p*-orbitals to lead *p*-orbitals allow electrons to move into the halogen bond. Therefore, exciting the intercalated band is very weak. We used a green cw laser (530 nm) to activate $[\text{H}_3\text{N}(\text{CH}_2)_8\text{NH}_3]\text{PbI}_4\cdot\text{I}_2$ at 4 K and no broadband emission was observed. This is in contrast to what was observed using the 415 nm laser, *vide infra*. However, using a high energy tuneable pulsed laser, weak emission from the broadband was observed with excitation up to 540 nm. Figure S27-S30 show the results from PL experiments.'

10. Text in the Conclusion section has been amended from 'Photoluminescence studies of the intercalated perovskite samples show a second, tuneable, broadband emission below 120 K, which is attributable to STEs, although intercalation modifies the nature of the STE to permit direct excitation into the intercalated band.' To 'Photoluminescence studies of the intercalated perovskite samples show a second, tuneable, broadband emission below 120 K, which is attributable to STEs, although intercalation modifies the nature of the STE to permit direct excitation into the intercalated band.'

Reviewer #1

Authors have addressed my comments, accept a few of them as given below-

Please add the method that was used for calculating the layer-shift factors (L_s).

We thank the reviewer for this suggestion and have included Figure R-1 as Figure S37 in the Supplementary Information (SI) to show how the calculation has been performed. For further information on existing use of the layer-shift factors, we also direct the reader to papers by Lightfoot (J. A. McNulty and P. Lightfoot, *IUCrJ*, 2021, 8, 485–513, DOI: 10.1107/S205225252100541), Marder (M-H Tremblay, J. Bacsa, B. Zhao, F. Pulvirenti, S. Barlow and S. R. Marder, *Chem. Mater.*, 2019, 31, 6145–6153, DOI: 10.1021/acs.chemmater.9b01564) and Tarasov (E. I. Marchenko, V. V. Korolev, A. Mitrofanov, S. A. Fateev, E. A. Goodilin and A. B. Tarasov, *Chem. Mater.*, 2021, 33, 1213–1217, DOI: 10.1021/acs.chemmater.0c03935).

The following text has been added to the SI:

“Layer shift factors (L_s) calculation

In order to calculate the layer shift factor, L_s , the crystal structure should be viewed along the out of plane direction, so that the long stacking axis is pointing towards you. The highest symmetry Ruddlesden-Popper or Dion-Jacobson phases adopt tetragonal symmetry. Many other related structures, including those reported in this paper adopt orthorhombic or monoclinic crystal symmetry, so L_s must be calculated using a slightly different method. For orthorhombic structures, as $\beta = 90^\circ$, the out-of-plane direction aligns with one of the unit cell axes, which has been denoted as c in this example (Figure R-1). For monoclinic crystal systems, where the $\beta \neq 90^\circ$, the projection process requires a rotation of $(\beta - 90)^\circ$ to align the out-of-plane direction.

Figure R-1: Diagram to illustrate the calculation of the L_s . Lead atoms in adjacent layers are represented by blue (layer 1) and pink (layer 2) spheres. The unit cell is outlined by the black solid square, and the in-plane Pb-Pb distance is indicated by black dashed lines.

As shown in Figure R-1, L_s is the ratio of the in-plane displacements of the Pb atoms in Layer 2 (shown in pink) relative to those in Layer 1 (shown in blue), along the two in-plane directions, which have been denoted as x and y . Mathematically, L_s is expressed as $(x'/x, y'/y)$, where x' and y' represent the projected in-plane distances of the atoms in Layer 2 relative to Layer 1 along the x and y directions.”

On page 4, the authors write, “In addition, we note that the single crystal structures published had high R-factors and large residual electron densities, which may indicate issues with the crystal structure or refinement. (ref. 14) (PEA-OH)PbBr₄·DMSO also could be used as a photodetector.” I do not agree with this statement as the single-crystal X-ray data and the PXRD data reported in this ref. 14 are of better quality than some of the compounds reported in this submission (Table S3). For example, the R-factor of [H₃N(CH₂)₇NH₃]PbBr₄·IBr is 12.5% and has high electron density, which is not modeled. R-factors of [H₃N(CH₂)₈NH₃]PbI₄·I₂ and [H₃N(CH₂)₉NH₃]PbI₄·I₂ are also high. So the authors should remove/reword this statement.

We thank the reviewer for noticing this and we have removed the statement. This section now reads as follows:

“Here, the intercalation of DMSO was very stable, due to hydrogen bonds between the PEA-OH and DMSO, enabling its use as a photodetector.¹⁴ However, the changes in electronic structure were small and (PEA-OH)PbBr₄·DMSO also had a short carrier lifetime.¹⁴ It was also possible to intercalate DMF into (PEA-OH)PbBr₄, and both (PEA-OH)PbBr₄·DMF and (PEA-OH)PbBr₄·2DMF were reported.¹⁴ Variable quantities of DMF could be intercalated, which led to mixed phase materials being observed.¹⁴”

I do agree with the authors that the unit cell is expected to increase in size as temperature is raised and it is obvious in most of the Rietveld refined PXRD data that they have provided in the revised work. However, I do not fully agree with the author's explanation about the mismatch of some of the PXRD data. For example, in the case of [H₃N(CH₂)₈NH₃]PbBr₄·I₂ (Figure S10), there are two intense peaks at 2θ ~ 15 ° and ~ 23 ° angles, which are absent in the simulated pattern. These peaks could be due to some impurity/secondary phase but cannot be due to the difference in temperature of single crystal and powder diffraction data. Similarly, in [H₃N(CH₂)₇NH₃]PbBr₄·IBr (Figure S8), there is an extra peak at ~ 20 ~ 7 °. So, some discussion on PXRD data needs to be added to the manuscript to indicate the mismatch. Also, please clarify why were the single crystal diffraction data collected in some cases at 173 K and others at 298 K.

Higher-quality PXRD data could not be obtained for [H₃N(CH₂)₈NH₃]PbBr₄·I₂ and [H₃N(CH₂)₇NH₃]PbBr₄·IBr, as these two intercalated layered perovskites tend to undergo partial de-intercalation when grinding to produce powder suitable for PXRD. However, this was not observed in intercalated perovskites where the X-site and intercalant are size-matched as de-intercalation did not occur when grinding [H₃N(CH₂)₇NH₃]PbBr₄·Br₂, even though it has the lowest thermal stability.

The PXRD data of [H₃N(CH₂)₇NH₃]PbBr₄·IBr do indeed show the presence of a small impurity phase, which has been identified as [H₃N(CH₂)₇NH₃]PbBr₄ and is responsible for the extra peak at 2θ ~ 7°. We note that in the literature to date, the structure of [H₃N(CH₂)₇NH₃]PbBr₄ has not been studied using single crystal XRD, due to the well-known instability of the *m*=7 phases. However, Liu et al. (C. Deng, G. Zhou, D. Chen, J. Zhao, Y. Wang, Q. Liu J. Phys. Chem. Lett. 2020, 11, 2934–2940 <https://doi.org/10.1021/acs.jpclett.0c00578>) built a structural model for [H₃N(CH₂)₇NH₃]PbBr₄, using the published structure of [H₃N(CH₂)₉NH₃]PbBr₄. They found that [H₃N(CH₂)₇NH₃]PbBr₄ could be indexed to space-group *Cc*, with unit cell parameters *a* = 25.070(3) Å, *b* = 8.1386(7) Å, *c* = 8.1104(9) Å and β = 90.85(9)°. This enabled a refinement to be carried out using powder X-ray diffraction data and a structure to be reported. As a result, the [H₃N(CH₂)₇NH₃]PbBr₄ phase was added to our refinement and Figure S8 has been updated, and the following discussion has been added in the SI:

“[H₃N(CH₂)₇NH₃]PbBr₄·IBr showed a small proportion of another phase in its PXRD pattern and this is modelled in the Pawley refinement as a small proportion of the parent phase, [H₃N(CH₂)₇NH₃]PbBr₄ (Figure S8).¹⁸”

We thank the reviewer for their comment on the PXRD data of [H₃N(CH₂)₈NH₃]PbBr₄·I₂. Due to equipment availability, we have been unable to get low-temperature PXRD data to match the data collection temperature of the SCXRD data, to rule out the possibility of phase transitions in the material.

As a result, we attempted to obtain the single-crystal structure of $[\text{H}_3\text{N}(\text{CH}_2)_8\text{NH}_3]\text{PbBr}_4\cdot\text{I}_2$ at room temperature. However, thin crystals and polycrystallinity in the sample caused problems in obtaining publication quality, room temperature SCXRD data. This was especially evident when trying to get reliable unit cell parameters for the long axis of the material. The best unit cell parameters were obtained in space group $C2$ unit cell parameters of $a \sim 8.26 \text{ \AA}$, $b \sim 8.28 \text{ \AA}$, $c \sim 30.40 \text{ \AA}$ and $\beta \sim 92.25^\circ$, although this structure was not good enough for publication. These unit cell parameters have been used in a Pawley refinement with the $[\text{H}_3\text{N}(\text{CH}_2)_8\text{NH}_3]\text{PbBr}_4\cdot\text{I}_2$ phase and there is good agreement between the experimental and calculated PXRD patterns (Figure S10). Further studies will focus on getting higher quality crystals, suitable for room temperature SCXRD structure determination.

The following discussion has been added to the discussion of the PXRD data in the SI:

“ $[\text{H}_3\text{N}(\text{CH}_2)_8\text{NH}_3]\text{PbBr}_4\cdot\text{I}_2$ also showed peaks associated with another phase, however these did not match the parent phase. Instead, they showed good agreement with a monoclinic C -centred phase that had been apparent during attempted SCXRD data collections on this compound at room temperature (Figure S10).”

In addition, further discussion was also given in the SCXRD experimental in the SI:

“Further data collection at ambient temperature was attempted on crystals of $[\text{H}_3\text{N}(\text{CH}_2)_8\text{NH}_3]\text{PbBr}_4\cdot\text{I}_2$. However, crystal quality problems meant that a dataset could not be collected at room temperature which would allow full structure refinement. The best unit cell parameters were obtained in space group $C2$, with $a = 8.26 \text{ \AA}$, $b = 8.28 \text{ \AA}$, $c = 30.40 \text{ \AA}$ and $\beta = 92.25^\circ$, which showed the perovskite framework of the structure, but the organic ammonium cation was disordered and the C/N positions could not be reliably determined. These unit cell parameters were, however, suitable to be used, in combination with those of the 173 K structure of $[\text{H}_3\text{N}(\text{CH}_2)_8\text{NH}_3]\text{PbBr}_4\cdot\text{I}_2$, in the Pawley fit to the PXRD data for this structure (Figure S10).”

For clarity, the following phrase was also added to the experimental section for $[\text{H}_3\text{N}(\text{CH}_2)_8\text{NH}_3]\text{PbBr}_4\cdot\text{I}_2$

“PXRD showed that at room temperature, the resulting dark-yellow powder contained $[\text{H}_3\text{N}(\text{CH}_2)_8\text{NH}_3]\text{PbBr}_4\cdot\text{I}_2$ as the major phase and a C -centred monoclinic phase as a secondary phase. Due to equipment availability, we have been unable to get low-temperature PXRD data to match the data collection temperature of the SCXRD data.”

Reviewer #2 (Remarks to the Author):

I am happy with the changes. The work can be published, in my opinion.

We thank the reviewer for their helpful suggestions and the time dedicated to reviewing our work.

Reviewer #3 (Remarks to the Author):

We thank the reviewer for their helpful suggestions and the time dedicated to reviewing our work.